# Perturbations in the microbiota-gut-brain axis shaped by social status loss
Ruijing Yang[1,2,3,4], Xin Wang [1,2,3,4], Jie Yang[1,2,3], Xingyu Zhou[1,2,3], Yiyuan Wu[1,2,3], Yifan Li[1,2,3], Yu Huang[1,2,3], Jianping Zhang[1,2,3], Ping Liu[1,2,3], Minghao Yuan[1,2,3], Xunmin Tan[1,2,3], Peng Zheng [1,2,3,5] ✉ & Jing Wu [1,2,3,5] ✉

Social status is closely linked to physiological and psychological states. Loss of social dominance can lead to brain disorders such as depression, but the underlying mechanisms remain unclear. The gut microbiota can sense stress and contribute to brain disorders via the microbiota-gut-brain axis (MGBA). Here, using a forced loss paradigm to demote dominant mice to subordinate ranks, we find that stress alters the composition and function of the gut microbiota, increasing *Muribaculaceae* abundance and enhancing butanoate metabolism, and gut microbial depletion resists forced loss-induced hierarchical demotion and behavioral alteration. Single-nucleus transcriptomic analysis of the prefrontal cortex (PFC) indicates that social status loss primarily affected interneurons, altering GABAergic synaptic transmission. Weighted gene co-expression network analysis (WGCNA) reveals modules linked to forced loss in the gut microbiota, colon, PFC, and PFC interneurons, suggesting changes in the PI3K-Akt signaling pathway and the glutamatergic synapse. Our findings provide evidence for MGBA perturbations induced by social status loss, offering potential intervention targets for related brain disorders.

Social animals frequently engage in social competition for resources such as territory and food, leading to the formation of hierarchical structures within their groups[1–5]. An individual's social dominance significantly influences its access to resources, thereby affecting its survival, health, reproduction, and other behaviors[6]. Subordinate rodents often exhibit anxiety-like behaviors, immunosuppression, elevated basal corticosterone levels, and shorter lifespans[7], while dominant mice tend to perform better in foraging and spatial learning memory[8,9]. Furthermore, losing dominant status or experiencing downward social mobility can markedly increase the risk of depression[10] and schizophrenia in humans[11,12]. Therefore, exploring the central neural mechanisms that influence social status is crucial.

Serving as a two-way regulatory pathway, the MGBA mediates communication between the gut microbiota and the central nervous system (CNS), essential for sustaining homeostasis in the gastrointestinal environment, microbiota, and central nervous system[13,14]. Previous researches have demonstrated that dysbiosis of the gut microbiota could influence the onset and progression of various neurological disorders, including autism, schizophrenia, depression, anxiety, Parkinson's disease, and Alzheimer's disease (AD)[15–17]. The gut microbiota is recognized as an important

contributor to stress-induced behavioral abnormalities. Chronic stress can induce dysbiosis of the gut microbiota, and prebiotic treatment can improve this dysbiosis and alleviate depression-like and anxiety-like behaviors[18]. Rodent models have shown that intestinal permeability increased during stress[19], and the increased intestinal permeability in depressive mice subjected to maternal separation stress can be reversed by antidepressant treatment[20]. Synbiotic treatment, which combines probiotics and prebiotics, can reduce the inflammatory response in the ileum and prefrontal cortex induced by chronic stress, promoting resilience to depression-like and anxiety-like behaviors in mice[21]. Wang et al. have also indicated that there were differences in the gut microbiota composition between dominant and subordinate individuals, and colonization with *Clostridium butyricum* can enable subordinate rats to regain dominance[22], suggesting that social hierarchy and associated stress could influence the gut microbiota. However, the deeper mechanisms of the MGBA, especially those related to the loss of dominant status, remain to be elucidated.

Social status is closely linked to the brain[23], and PFC is recognized as the essential brain area controlling social hierarchal behavior[24,25]. Changes in social status are mediated by synaptic strength in the mediodorsal thalamus-dorsal

[1]Department of Neurology, The First Affiliated Hospital of Chongqing Medical University, Chongqing, China. [2]Key Laboratory of Major Brain Disease and Aging Research (Ministry of Education), Chongqing Medical University, Chongqing, China. [3]Institute for Brain Science and Disease, Chongqing Medical University, Chongqing, China. [4]These authors contributed equally: Ruijing Yang, Xin Wang. [5]These authors jointly supervised this work: Peng Zheng, Jing Wu. ✉e-mail: pengzheng_cqmu@yeah.net; jing-wu@hospital.cqmu.edu.cn

medial prefrontal cortex (dmPFC) pathway[26]. Fan et al. reported that forced loss activated the lateral habenula, which enhances retreat behavior in mice during tube tests by inhibiting the medial prefrontal cortex[27]. While current research primarily focuses on the neural circuit mechanisms underlying social status[28,29], a study also highlights the critical role of neuron-glia interactions in social hierarchy[30]. Given the heterogeneity and functions of neuronal cell types, Newton et al. demonstrated that single-nucleus RNA sequencing (snRNA-seq) could identify transcriptome changes in specific cell types related to depressive behaviors induced by the chronic unpredictable mild stress model[31]. Therefore, snRNA-seq technology enables us to investigate cellular heterogeneity in the CNS, identify transcriptome alterations in cell types associated with forced loss-related behaviors, and provide insights into the neurological changes induced by social status loss.

Here, we assessed the social hierarchy among mice using the tube test and induced a reduction in their dominance rank through a forced-loss paradigm. Based on this model, we conducted 16S ribosomal RNA sequencing (16S rRNA-seq) and metagenomic analyses on fecal samples and colonic contents to evaluate whether forced loss led to alterations in gut

microbiota composition and function. Besides, we administered broad-spectrum antibiotics to deplete gut microbiota, thereby validating its regulatory influence. Subsequently, we performed RNA sequencing (RNA-seq) on colonic tissues to identify changes in the intestinal transcriptome. Furthermore, we utilized snRNA-seq to analyze cell-type specific expression, aiming to identify the cell types highly correlated with forced-loss behaviors. Finally, WGCNA was applied to identify key gene modules associated with social status loss in the gut microbiota, colon, and PFC, revealing disruptions in the PI3K-Akt signaling pathway and the glutamatergic synapse within the MGBA induced by social status loss.

## Results

### Forced loss significantly altered the social hierarchy and behaviors of mice

First, we utilized the tube test, a common behavioral paradigm, to measure social hierarchy in mice[32], which can divide 4 mice in a cage into 4 ranks (Fig. 1a). These mice maintained a relatively stable rank throughout the experimental period (Fig. 1b). Notably, body weight and movement

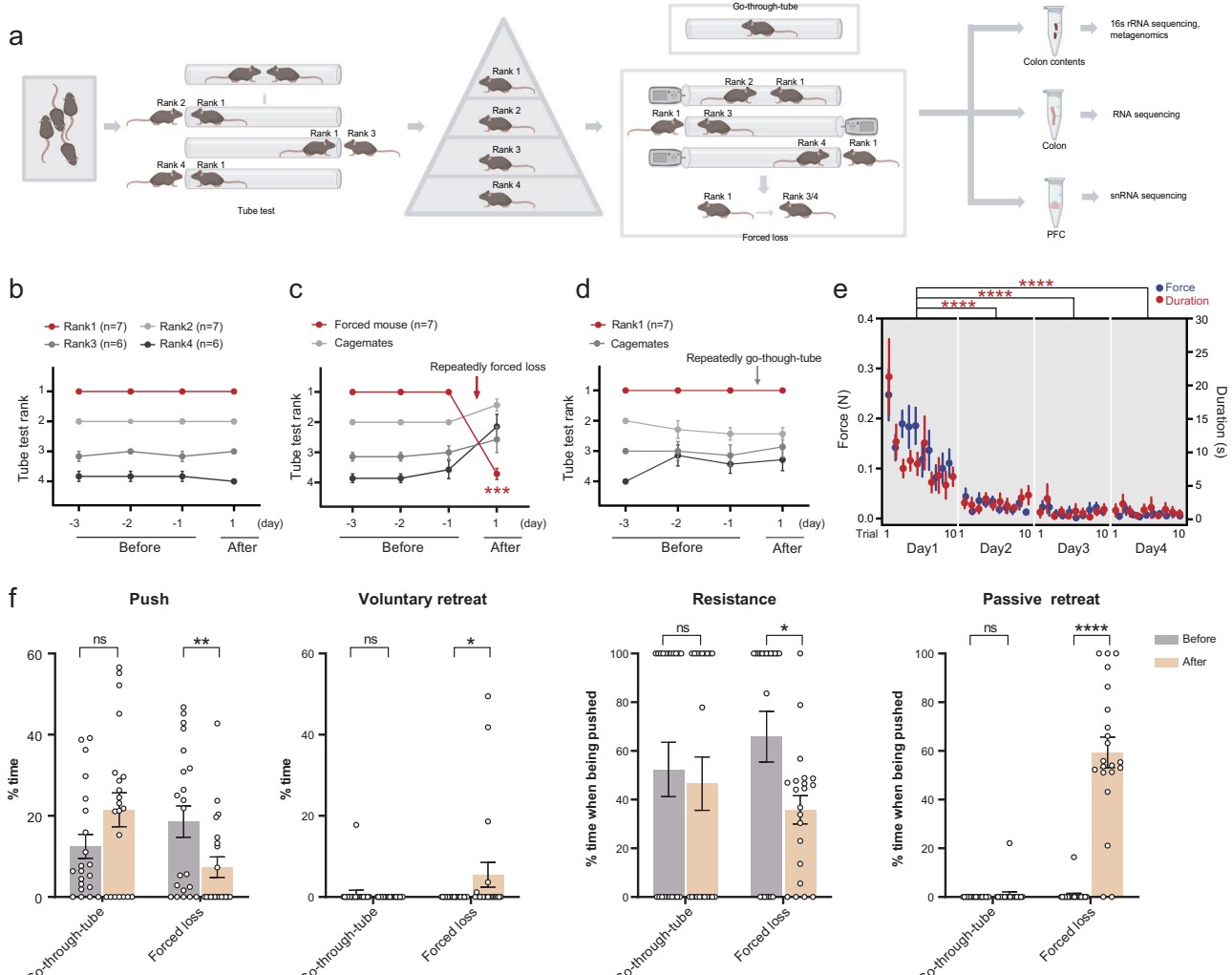

**Fig. 1 | Forced loss altered social hierarchy and behaviors in mice. a** Schematic diagram showed the process of obtaining mice of various ranks via the tube test, establishing forced loss and control groups among rank1 mice through forced loss or go-through-tube procedure, followed by sample sequencing. The illustration was created in BioRender. Yang, R. (2025) https://BioRender.com/c35i595. **b–d** Daily tube test outcomes for mice with relatively stable ranks, rank1 mice before and after forced loss, and rank1 mice before and after go-through-tube (***$p = 0.0006$, Mann–Whitney test, two-tailed). **e** Average force (colored blue) and duration

(colored red) for individual trials throughout the period of forced loss (****$p < 0.0001$, Mann–Whitney test, two-tailed). **f** Time percentage of pushing, voluntary retreating, resistance (when pushed by an opponent), and passive retreating (when pushed by an opponent) during tube test for mice before and after go-through-tube or forced-loss procedure (Go-through-tube, $n = 7$; Forced loss, $n = 7$; 21 trials per group). *$p < 0.05$, **$p < 0.01$, ***$p < 0.001$, ****$p < 0.0001$, ns, not significant; data are mean ± SEM; Mann–Whitney test, two-tailed).

distance did not exhibit significant differences among mice of different ranks, indicating that the formation of social hierarchy was not influenced by body weight or locomotor ability (Supplementary Fig. 1a). Rank1 mice were the dominant individuals, and the rest were subordinate. We then used a forced-loss paradigm to compel the dominant mouse to be demoted from its rank, thereby stripping it of its dominant status[27] (Fig. 1a). Therefore, the forced-loss mice were those that lost their dominant status after being defeated by subordinate mice in the tube test (forced-loss group). As a control, the go-through-tube mice merely passed through the tube during the tube test without encountering resistance from subordinate mice, thus maintaining their rank1 status (go-through-tube group).

In our study, we found that the behaviors of forced-loss mice differed significantly from those of dominant mice. Forced loss mice experienced a marked decline in rank after repeating forced loss (Fig. 1c), while mice in the go-through-tube group maintained their rank1 status (Fig. 1d). To investigate whether dominant mice exhibited resistance and how the resistance changed during the forced-loss process, we connected a dynamometer to the tube blocker on the subordinate mouse's side to indirectly assess the force exerted by the original rank1 mice when pushing against the subordinate mice. We found that the force and duration exerted by the original rank 1 mice on the subordinate mice gradually decreased over time (Fig. 1e). This indicated that the resistance of the original rank 1 mice diminished, making them more susceptible to losing to the subordinate mice.

Furthermore, to ascertain whether the behavior of dominant mice changed before and after forced loss, we analyzed their behavior in the tube test. Before and after the intervention, we found that go-through-tube mice showed no significant changes in behaviors during the tube test, while forced-loss mice exhibited reduced pushing, increased voluntary retreating, and decreased resistance with increased passive retreating when being pushed (Fig. 1f). These findings suggest that the behavior manifestation of dominant mice in the tube test was significantly altered following forced loss. Open-field tests and body weight measurements revealed no significant differences in body weight and locomotor ability between the forced loss and the go-through-tube group, suggesting that forced loss did not affect these factors (Supplementary Fig. 1d). These findings report that forced loss significantly alters the rank of dominant mice and their behaviors in tube test, while go-through-tube mice maintain stable rank and behaviors.

### Forced loss led to alterations in gut microbiota in mice

To explore the impact of social hierarchy on gut microbiota, 16S rRNA-seq was employed to assess alterations in the gut microbiota (Fig. 1a). Our results indicated that social status did not influence the diversity, richness, or composition of the gut microbiota in mice (Supplementary Fig. 1b, c). However, forced loss led to alterations in the gut microbiota. α-diversity analysis revealed no significant differences in gut microbial diversity and richness between the forced loss and the go-through-tube group (Supplementary Fig. 1e). However, β-diversity analysis using principal coordinates analysis (PCoA) demonstrated significant differences in community structure at the genus level across the two groups (Fig. 2a), with the relative abundance of gut microbiota at the genus level depicted in Fig. 2b. Species difference analysis at the ASV level identified 16 differential ASVs between the forced loss and the go-through-tube group. Notably, forced-loss mice were characterized by 8 elevated ASVs, primarily attributed to *Muribaculaceae* (4 ASVs). Furthermore, correlation analysis indicated that 11 ASVs were significantly associated with forced loss-related behaviors, particularly voluntary retreat, resistance, and passive retreat (Fig. 2c).

To elucidate the interaction relationships within the gut microbiota, we established a co-expression network founded on microbial abundance. The differential ASVs automatically assembled into two distinct clusters, with ASVs from each cluster predominantly enriched in either the forced loss or the go-through-tube group. It was observed that ASVs within both cluster 1 and cluster 2 were positively correlated within their respective clusters, and 4 ASVs in cluster 2 belonged to *Muribaculaceae*. Differently, some ASVs from

cluster 1 were negatively associated with those in cluster 2 (Fig. 2d), suggesting potential synergistic effects among these differential ASVs in forced-loss mice.

To investigate the functional alteration of the gut microbiota resulting from forced loss, we conducted metagenomic analysis (Fig. 1a). Linear discriminant analysis effect size (LEfSe) analysis of Kyoto Encyclopedia of Genes and Genomes (KEGG) functions (LDA > 1.3[33]) highlighted functional differences between the forced loss and the go-through-tube group, primarily involving butanoate metabolism (Fig. 2e). Butanoate metabolism, also known as butyrate metabolism, is a pathway that produces short-chain fatty acids (SCFAs), which are important signaling molecules in the MGBA. Butyrate, one of the primary SCFAs, has been shown to have multiple beneficial effects on both gut health and brain function[13]. The co-expression network analysis indicated a significant positive correlation between butanoate metabolism, valine, leucine, and isoleucine degradation, and lysine degradation, all of which were enriched in the forced-loss group. In contrast, the nonribosomal peptide structures, retrograde endocannabinoid signaling, and regulation of lipolysis in adipocyte functions, which were reduced in the forced-loss group, also showed a significant positive correlation among themselves, with the latter two belonging to organismal systems (Fig. 2f). These findings suggest that forced loss primarily affected butanoate metabolism in the gut microbiota.

### Gut microbiota depletion resisted forced loss-induced hierarchical and behavioral alteration in mice

Investigating the role of gut microbiota in forced loss-induced hierarchical demotion and behavioral alteration, we compared microbiota-intact mice with microbiota-depleted mice. The microbiota-depleted mice were generated through intragastric administration of broad-spectrum antibiotics (ABX) prior to forced loss and go-through-tube paradigms, with quantitative PCR (qPCR) confirming effective microbial depletion (Fig. 3a and Supplementary Fig. 1f). Compared to microbiota-intact mice, microbiota-depleted mice exhibited marked resistance to hierarchical demotion after forced loss (Fig. 3b). While 76.92% of dominant (rank1) microbiota-intact mice were demoted to subordinate status following forced loss, only 28.57% of microbiota-depleted mice lost dominance status. After the forced loss, the proportion of rank1 mice retaining dominance increased from 7.69% in microbiota-intact group to 42.86% in the microbiota-depleted group (Fig. 3c). These results demonstrate that gut microbiota regulates forced loss-driven hierarchical demotion.

Force and behavior analyses reveal that gut microbiota depletion enhances resistance to forced loss in dominant mice. Dynamometer-based force quantification found that dominant mice with depleted microbiota gradually exerted less force against subordinate mice over time, while the duration of the resistance remained unchanged (Fig. 3d). Notably, during the day 2 to 4 of the forced-loss process, microbiota-depleted dominant mice displayed elevated force and prolonged duration toward subordinates compared to microbiota-intact dominant mice (Fig. 3e). Besides, microbiota-depleted mice predominantly exhibited pushing and resistance rather than retreating, and microbiota-depleted mice exhibited stable behaviors before and after forced loss, with no significant difference in pushing, voluntary retreating, or resistance behaviors (Supplementary Fig. 1g). Importantly, compared to microbiota-intact mice, microbiota-depleted mice exhibited increased pushing, reduced voluntary and passive retreating after forced loss (Fig. 3f), indicating gut microbiota depletion plays a role in forced loss-induced behavioral alteration. Collectively, these findings indicate that gut microbiota modulates forced loss-induced stress responses, whereas its depletion allows mice to retain dominance, resisting both hierarchical demotion and behavioral alteration induced by forced loss.

### Single-nucleus RNA-seq analysis in PFC

The nuclear sequencing process of PFC from the forced-loss group and the go-through-tube group is shown in Fig. 1a. After quality control, we captured a total of 52,626 nuclei and identified 21,042 genes from the PFC in the

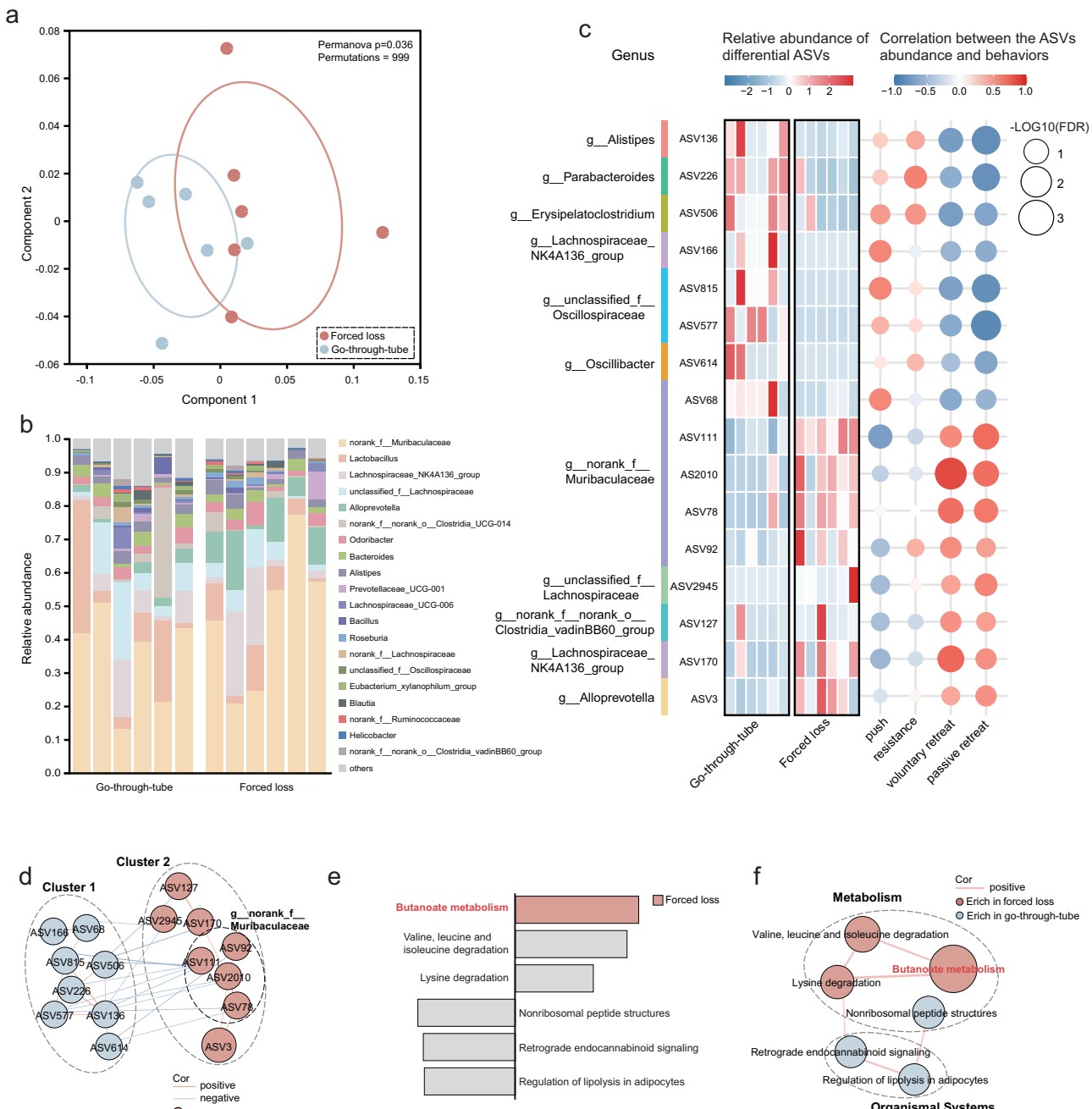

**Fig. 2 | Alterations in the composition and function of the gut microbiota in mice. a** The overall characteristics of gut microbiota were analyzed via PCoA utilizing the abund_jaccard distance, and internal discrepancies were evaluated via PERMANOVA test (Go-through-tube, $n = 6$; Forced loss, $n = 6$; permutations = 999, *$p = 0.036$). **b** Relative abundance of gut microbiota at the genus level. **c** Heatmap showed the relative abundance of differential ASVs across groups and their Spearman correlation with behaviors related to forced loss, with genus names of ASVs annotated (Wilcoxon rank-sum test, two-tailed). **d** Co-expression network among differential ASVs. Node size represents the abundance of the ASVs, and lines

connecting the nodes signify positive (red) or negative (blue) correlations ($r > 0.58$ or $< -0.58$, and $p < 0.05$; two-tailed Spearman correlation). **e** LEfSe analysis illustrated the impact of differential functions contributing to inter-group differences, with LDA scores > 1.3. The gray columns represented functions with LDA scores < 2, and only butanoate metabolism with LDA score > 2 was enriched in the forced-loss group (Go-through-tube, $n = 6$; Forced loss, $n = 6$; Wilcoxon rank-sum test, two-tailed). **f** Co-expression network among differential functions. Node size represents the abundance of the functions, and edges between nodes indicate positive correlation (red) ($r > 0.6$ or $< -0.6$ and $p < 0.05$; two-tailed Spearman correlation).

6 mice (3 in the forced-loss group and 3 in go-through-tube group). After dimensionality reduction and unsupervised graph-based clustering, 29 preprocessed clusters were identified in the PFC (Supplementary Fig. 2a). We annotated these clusters with main cell types based on previous summaries and reported marker genes from the literature[34,35]. These included excitatory neurons ($n = 32,822$ in the PFC; marked by *Slc17a7*, *Rorb*, *Cux2*), interneurons ($n = 8453$; *Gad1*, *Gad2*, *Lamp5*), microglia

($n = 1193$; *Ctss*, *C1qa*, *Hexb*), astrocytes ($n = 4570$; *Slc1a2*, *Slc1a3*, *Gja1*), oligodendrocytes ($n = 2779$; *Mbp*, *Mog*, *Mag*), Oligodendrocyte precursor cells (OPC; $n = 1161$; *Vcan*, *Neu4*, *Pdgfra*), endothelial cells ($n = 744$; *Rgs5*, *Igfbp7*) and fibroblasts ($n = 665$; *Mgp*, *Col1a*, *Ptgds*) (Fig. 4a–c). Compared with the go-through-tube group, there was no proportional difference in main cell types in the forced-loss group (Fig. 4d and Supplementary Fig. 2b).

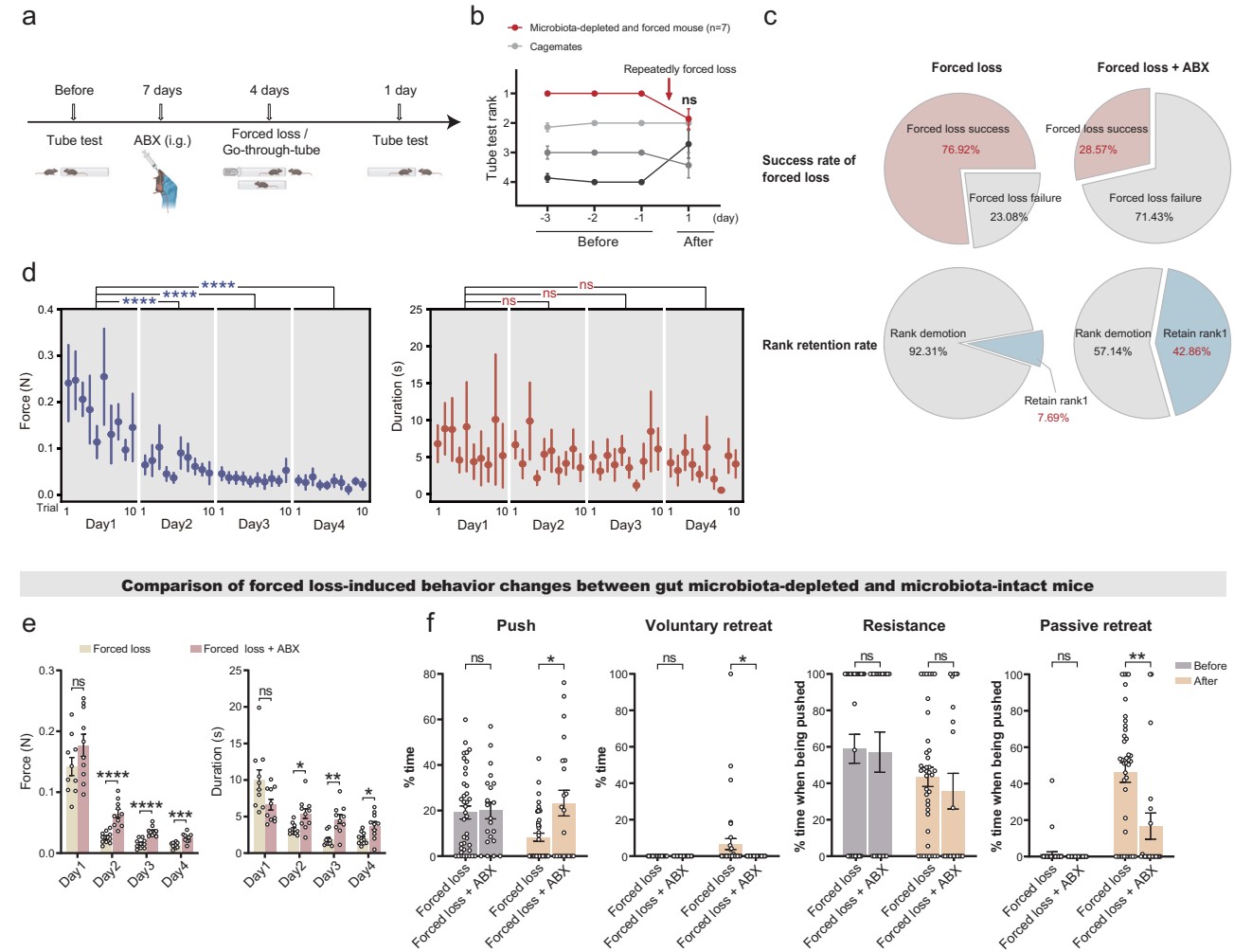

**Fig. 3 | Gut microbiota depletion regulated hierarchical and behavioral alteration induced by forced loss. a** Schematic of the experimental timeline: Antibiotic-mediated gut microbiota depletion (ABX) (Forced loss + ABX, $n = 7$; Go-through-tube + ABX, $n = 6$). The illustration was created in BioRender. Yang, R. (2025) https://BioRender.com/c35i595. **b** Daily tube test outcomes in microbiota-depleted mice before and after forced loss (Mann–Whitney test, two-tailed). **c** Success rates of demoting dominant mice to subordinate status and retaining rank1 in microbiota-depleted versus microbiota-intact mice after forced loss (Forced loss, $n = 13$; Forced loss + ABX, $n = 7$). Statistical significance determined by chi-square test (Success rate of forced loss: $\chi^2 = 2.105$, $p = 0.0353$; Rank retention rate: $\chi^2 = 3.516$, $p = 0.0608$). **d** Average force (blue) and duration (red) per trial during forced loss in microbiota-

depleted mice ($n = 7$ mice, 21 trials, Mann–Whitney test, two-tailed). **e** Comparison of average force and duration per trial during forced loss between microbiota-depleted and microbiota-intact mice (Forced loss, $n = 13$ mice, 10 trials / mice; Forced loss + ABX, $n = 7$ mice, 10 trials / mice; Mann–Whitney test, two-tailed). **f** Time percentage for pushing, voluntary retreating, resistance (when pushed by an opponent), and passive retreating (when pushed by an opponent) during tube tests in microbiota-depleted versus microbiota-intact mice before and after forced loss (Forced loss, $n = 13$ mice, 39 trials; Forced loss + ABX, $n = 7$ mice, 21 trials; Mann–Whitney test, two-tailed). Data are presented as mean ± SEM. \*$p < 0.05$, \*\*$p < 0.01$, \*\*\*$p < 0.001$, \*\*\*\*$p < 0.0001$; ns, not significant.

## Forced loss led to cell-specific transcriptome changes

To further elucidate the cell-specific transcriptomic alterations regulated by forced loss, we compared differentially expressed genes (DEGs) between the forced-loss group and the go-through-tube group in the PFC. Compared with the go-through-tube group, 812 DEGs were detected in the forced-loss group across the six major cell types, including 534 up-regulated and 278 down-regulated genes (Fig. 5a). Neurons contributed 55.91% of the DEGs, with a higher number of DEGs compared to glial cells (Supplementary Fig. 2c). The changes in gene expression were specific to each cell type, sharing only 83 DEGs across more than one cell type. The highest number of DEGs was detected in interneurons, including 333 genes that were exclusively found in interneurons. In contrast, the lowest number of DEGs, only 9, was identified in OPC (Fig. 5b, d and Supplementary Fig. 2d). Using the downsampling method to equalize the number of nuclei across the six major cell types, we conducted 10 comparisons[36] and consistently found the highest number of DEGs in interneurons. Compared with other cell types, interneurons exhibited prominent changes, suggesting that they might be

preferentially affected by forced loss (Fig. 5c). Furthermore, the Gene Ontology (GO) functions associated with the DEGs in each cell type differed (Fig. 5e). For example, the DEGs in interneurons were mainly related to GABAergic synaptic transmission, including *Car2, Cckbr, Erbb4, Grik1, Tac1*. The DEGs of excitatory neurons were enriched in synaptic function and biological metabolic processes, such as *Apoe, Nrg1, Robo1*. Astrocytes and oligodendrocytes cell-specific DEGs were primarily associated with synaptic function. Changes in microglia were mainly related to inflammation and immune regulation. The altered OPC-specific DEGs were enriched in signaling pathways and lipid metabolism (Supplementary Fig. 2e).

In order to explore potential correlations among cell-types, we constructed a co-expression network. Here, 41 DEGs were generated in three gene clusters (GC1–3) based on their high correlations. GC1, composed of genes from interneurons, microglia, and astrocytes, contributed to the modulation of signaling pathways and protein functions. Genes such as *Il1rapl1, Dlg2, Kcnd2*, and *Kcnb2* from astrocytes and interneurons, as well as *Cadm2, Plcb1*, and *Map2* specifically from interneurons, were primarily

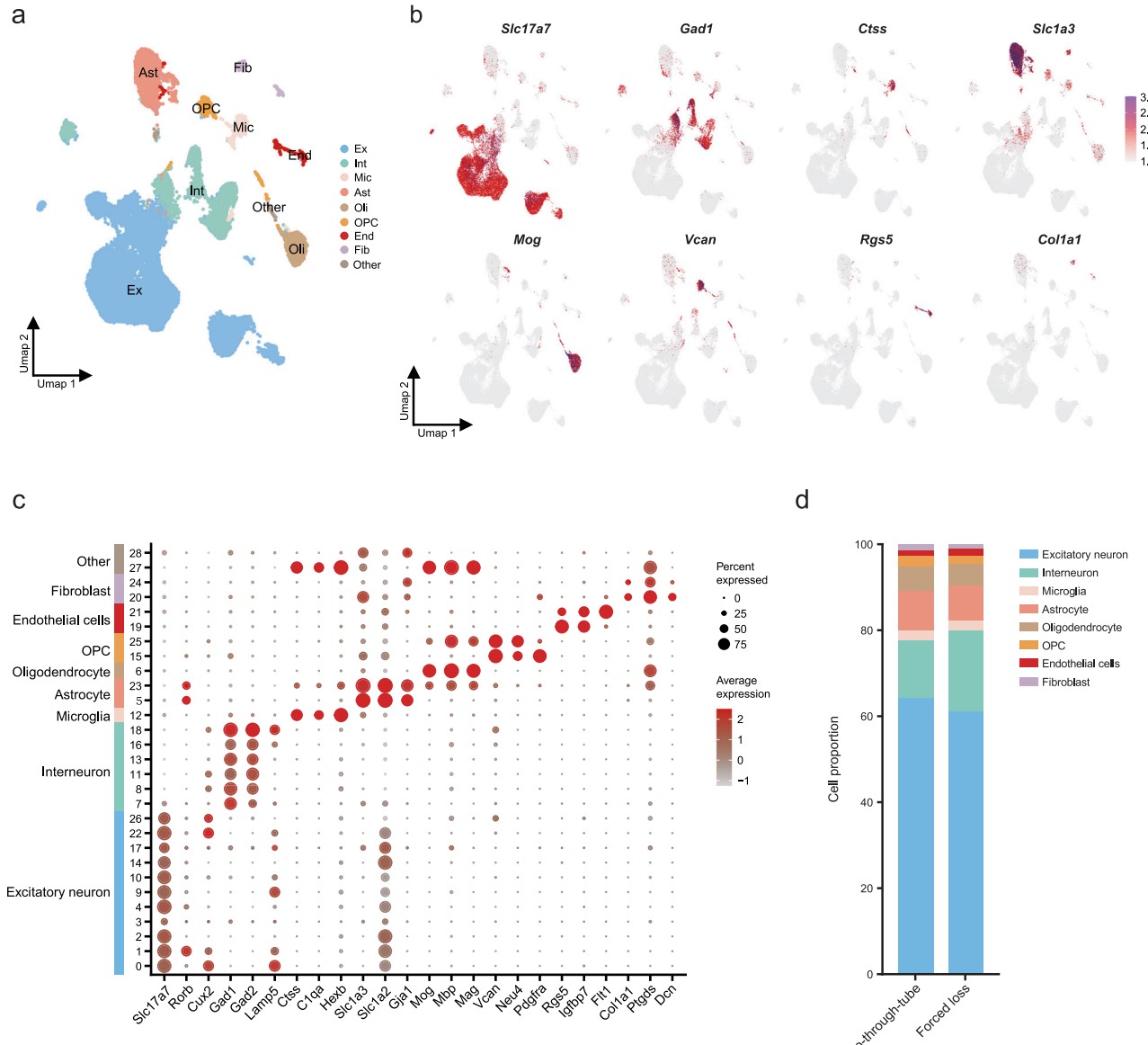

**Fig. 4 | Identification of cell types and cell proportion in PFC. a** The UMAP graph showed the clusters after cell annotation in PFC. **b** The most specific marker genes for the eight cell types in PFC. **c** Bubble plot illustrating marker genes expression in PFC. The diameter of each bubble illustrates the percentage of cells expressing their marker genes within the clusters (Pct. exp.), and the color intensity signifies the average level of gene expression (Avg. exp. scale). **d** Proportion of cell types in PFC (Go-through-tube, $n = 3$; Forced loss, $n = 3$).

involved in synaptic and neuronal functions. Notably, the genes *Il1rapl1*[37], *Kcnd2*[38], and *Map2*[39] are closely associated with the occurrence of brain diseases. The interaction between interneurons and *Rims1*, *Atp2b1*, and *Celf2* genes from glial cells involved in regulating synaptic vesicles and calcium ion transport were represented in GC2. In contrast, GC3 was expressed in several major cell types, indicating an interaction between neurons and glial cells (Fig. 5f).

Furthermore, CellChat[40] was used to compare the ligand-receptor pairs between the forced-loss group and the go-through-tube group, aiming to understand the effects of forced loss on the communication between interneurons and other major cell types. Within main cell types, 737 and 653 ligand-receptor pairs were recognized in the go-through-tube and the forced-loss group (Supplementary Fig. 3a). We then compared specific ligand-receptor pairs associated with interneurons, resulting 75 pairs in the go-through-tube group and 24 pairs in the forced-loss group. Interneurons primarily communicated with excitatory neurons, followed by oligodendrocyte precursor cells. Compared to the go-through-tube group, forced loss resulted in the greatest reduction in communication between interneurons

and excitatory neurons, decreasing from 46 pairs to 8 pairs (Fig. 5g, h). Additionally, the ligand-receptor relationships between interneurons and excitatory neurons were related to synaptic transmission and neural development (Supplementary Fig. 3b). For example, neurexins (*Nrxns*) play an essential role in neurotransmission and synaptic differentiation, and mutations in neurexin genes were associated with brain disorders such as autism and schizophrenia[41]. The reduced intercellular communication between interneurons and excitatory neurons indicates primarily changes in neuronal and synaptic functions.

## WGCNA showed potential associations between interneurons and forced loss-related behaviors

To analyze the cellular and gene modules that drive different phenotypes, we employed WGCNA and adopted four behaviors (push, voluntary retreat, resistance, passive retreat) with significant differential alterations. 126 modules were identified in six cell types (astrocytes, 24; oligodendrocytes, 23; interneurons, 20; excitatory neurons, 20; OPC, 20; microglia, 19) (Fig. 6a and Supplementary Fig. 4a–e). Voluntary retreat behavior was significantly

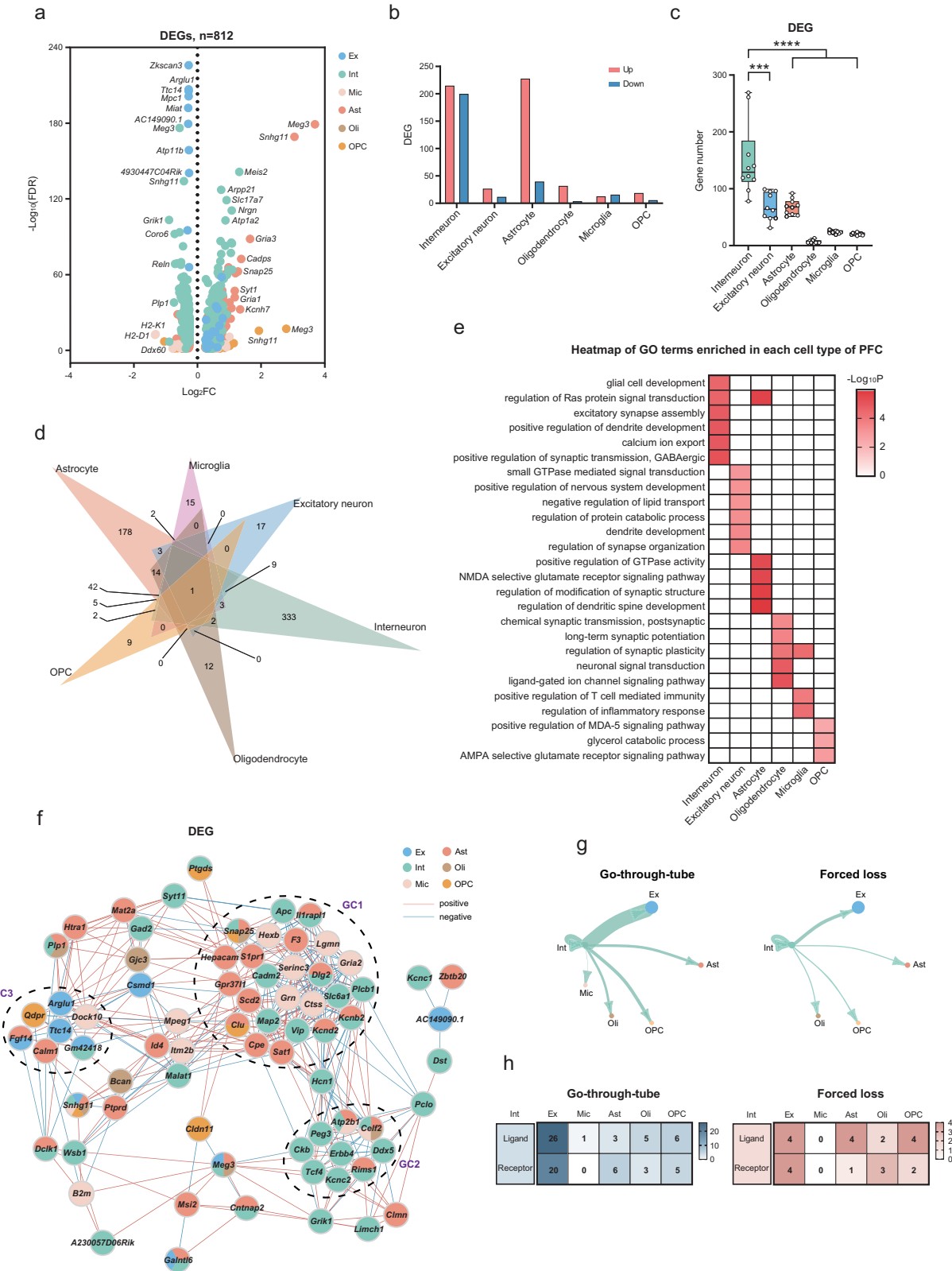

**Fig. 5 | Forced loss led to cell-specific transcriptome changes. a** Volcano plots depicted DEGs that were specific to different cell types (|log₂(FC)| > 0.25, and FDR < 0.05). **b** The number of DEGs up-regulated and down-regulated in six major cell types. **c** The downsampling analysis of DEGs demonstrated that interneurons exhibited a greater number of gene changes compared to other major cell types, with ten repetitions (Interneuron vs Excitatory neuron, ***$p$ = 0.0002; Interneuron vs Astrocyte, ****$p$ < 0.0001; Interneuron vs Oligodendrocyte, ****$p$ < 0.0001; Interneuron vs. Microglia, ****$p$ < 0.0001; Interneuron vs OPC, ****$p$ < 0.0001; Mann–Whitney test, two-tailed). **d** The Venn diagram showed the distribution of

812 DEGs specific to each of the six major cell types within PFC. **e** Heatmap showed typical functional pathways enriched by cell-specific DEG in six major cell types. **f** Co-expression network showed three gene clusters (GC1–3) formed by clustering ($r$ > 0.75 or < −0.75, and $p$ < 0.05; two-tailed Spearman correlation). **g** Ligand-receptor pairs in the go-through-tube and forced-loss groups. The size of the circles and the thickness of the lines are indicative of the number of ligand-receptor pairs included. **h** The amount of ligand-receptor pairs that were activated specifically in the go-through-tube and forced-loss groups.

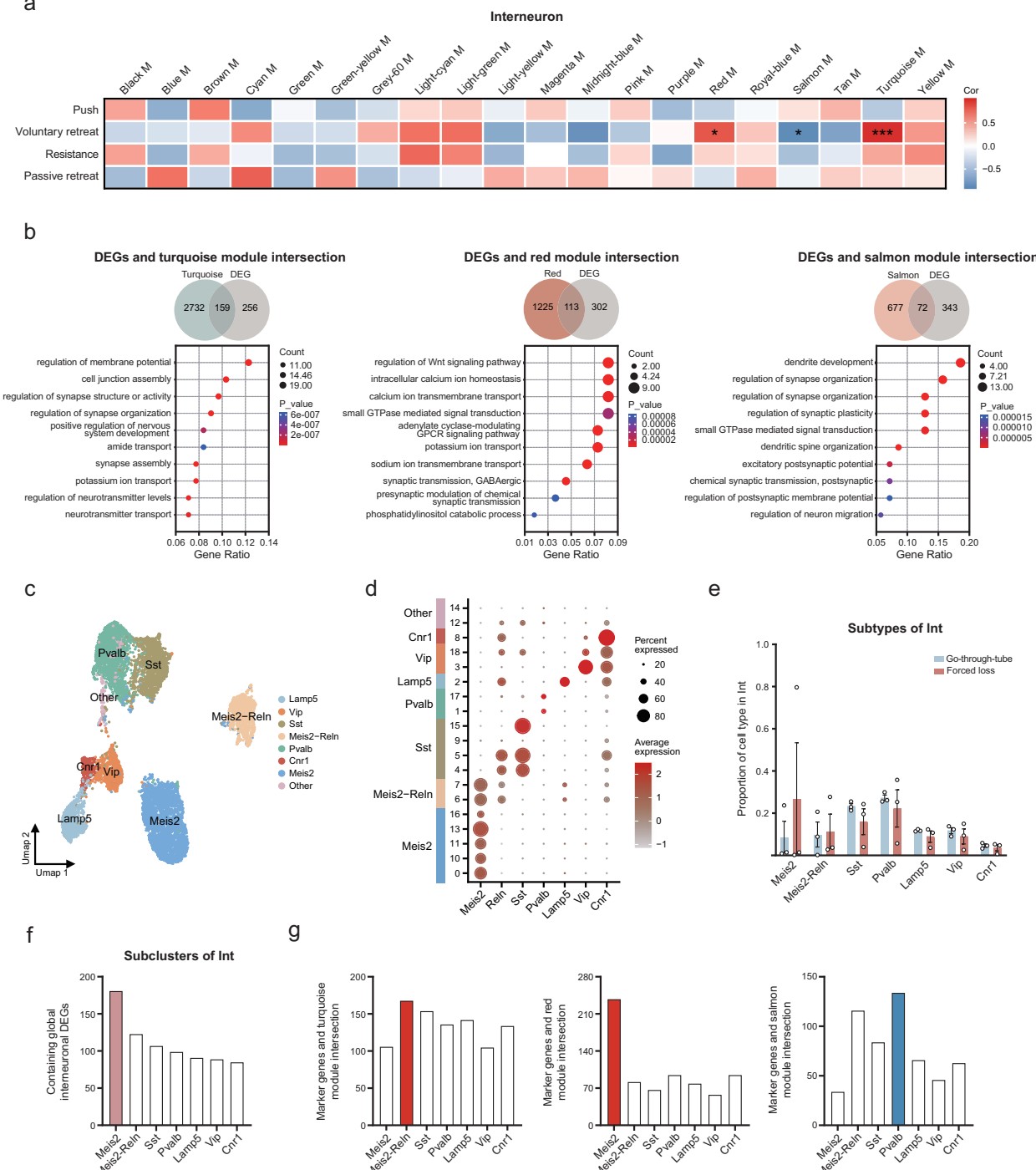

**Fig. 6 | The potential associations between interneurons and forced loss-related behaviors. a** The heatmap showed gene modules associated with four behaviors (push, voluntary retreat, resistance, passive retreat) in interneurons ($r > 0.6$ or $< -0.6$, $*p < 0.05$, $**p < 0.01$, $***p < 0.001$, two-tailed Pearson correlation). **b** Venn diagram illustrated the overlap of genes in three modules (turquoise, red, salmon) with interneuron DEGs. GO enrichment analysis showed the functions of genes enrichment at the intersection of three modules and DEGs ($p < 0.05$). **c** The UMAP graph of the major cell subtypes in interneuron. **d** Bubble plot of marker genes expression in interneuron subtypes. **e** Proportion of cell subtypes in interneuron (Go-through-tube, $n = 3$; Forced loss, $n = 3$; Go-through-tube vs. Forced loss, no statistical difference in $p$-values; data are mean ± SEM; Mann–Whitney test, two-tailed). **f** Major contributions of Meis2 subtype to global interneuron DEGs. **g** The association between the marker genes of interneuron subtypes and the gene modules related to voluntary retreat behavior.

associated with the largest number of gene modules in four forced loss-related behaviors, suggesting that this behavior was more vulnerable to regulation by gene expression. Compared to other cell types, interneurons contained the greatest number of gene modules relevant to voluntary retreat behavior (three modules). Interestingly, all three behavior-related gene modules in interneurons were only associated with voluntary retreat. In addition, these three gene modules exhibited significant overlap with the interneuron DEGs (Fig. 6b, $p = 1.85e\text{-}36$, 159 genes in turquoise M; 1.2e-41, 113 genes in red M; 1.72e-29, 72 genes in salmon M; hypergeometric test). Therefore, we explored the biological functions related to voluntary retreat behavior in interneurons (Supplementary Fig. 4f). Two gene modules (turquoise M and red M) showed a positive correlation with voluntary

retreat behavior, including genes such as *Adgrb1* and *Akap7*, which were involved in ion transmembrane transport, signaling pathways, and synaptic transmission. In contrast, the negatively correlated salmon module, containing genes such as *Arhgap33* and *Baiap2* genes, was primarily associated with dendrite and synaptic functions. These findings indicate that alterations in synaptic function in interneurons may drive the voluntary retreat behavior in forced-loss mice.

### Contribution of interneuron subtypes to forced loss-related gene modules

We further analyzed whether the interneuron subtypes were altered. All 8453 interneurons were clustered into seven subtypes: Meis2, Meis2-Reln, Sst, Pvalb, Lamp5, Vip, Cnr1 (Fig. 6c, d). We found no difference in the proportion of interneuron subtypes between the forced-loss group and the go-through-tube group (Fig. 6e). In addition, 181 Meis2 subtype gene markers overlapped with the DEGs of interneurons, which was greater than in the other six subpopulations (Fig. 6f). Genes such as *Abl2* can regulate actin cytoskeleton organization and presynaptic axon guidance[42]. The Meis2 subtype gene markers, including *Celf2* and *R3hdm1*, are related to mRNA splicing and neuronal development[43-45]. To investigate the contribution of interneuron subtypes to forced loss-related gene modules, we further analyzed the association between the gene markers of interneuron subtypes and the gene modules related to voluntary retreat behavior (Fig. 6g). The turquoise and red modules, which were positively correlated with voluntary retreat behavior, showed the most significant overlap with the Meis2-Reln and Meis2 subtypes, involving ion transport and axonogenesis in neurons. The negatively correlated salmon module mainly overlapped with the Pvalb subtype, and the overlapping genes, such as *Slit2* and *Robo1*, were involved in cell growth and migration[46]. These results suggest that the Meis2, Meis2-Reln, and Pvalb subtypes may collectively influence forced loss-related behaviors.

### WGCNA analysis revealed MGBA signaling pathways associated with forced loss

Given that forced loss may lead to changes in the function and composition of the gut microbiota, we further investigated the colon, where these microbes reside, for associated alterations. We performed RNA-seq analysis on colonic tissue and identified DEGs, with 6 up-regulated and 6 down-regulated in the forced-loss group (Fig. 7a). GO analysis showed that these DEGs were involved in gut microbiota-host signaling, including positive regulation of the Notch signaling pathway and cysteine biosynthetic processes (Fig. 7b). Additionally, KEGG pathway analysis indicated that these DEGs could activate taurine and hypotaurine metabolism, glutathione metabolism, and arachidonic acid metabolism (Fig. 7c).

To further explore the potential regulatory pathways of the MGBA associated with forced loss, we compared gene modules from the gut microbiota, colon, PFC, and PFC interneurons. We found that gene modules associated with forced loss-related behaviors were present in all four datasets (Fig. 7d). The overlapped biological pathways across different tissue-derived gene modules suggested a coordinated role of the MGBA in regulating behaviors associated with forced loss. We found that the common pathways shared by the gut microbiota, colon, PFC, and PFC interneurons were the PI3K-Akt signaling pathway and the glutamatergic synapse. The modules to which the two pathways belong were significantly associated with the four forced loss-related behaviors: pushing, voluntary retreating, resistance, and passive retreating. The gene *Igf1*, which is associated with the PI3K-Akt signaling pathway, is crucial for the adult brain and promotes neurogenesis following trauma[47,48]. Besides, upregulation of the IGF1R-PI3K-AKT pathway can enhance colonic epithelial integrity and regeneration[49]. The overlapped pathways between the gut microbiota and PFC interneurons included ABC transporters and the TGF-beta signaling pathway, and the modules that these pathways belong to were significantly associated with pushing voluntary retreating, and resistance behaviors. Additionally, the modules containing the pathways shared by the gut microbiota and the PFC exhibit a significant correlation with the same three

forced loss-related behaviors. In addition, the modules containing the pathways shared by the gut microbiota and the colon were significantly associated with four forced loss-related behaviors (Fig. 7e). The shared pathways provide evidence that intercellular signal transmission, particularly through signaling pathways and neurotransmitter systems, regulates the effects of forced loss on mouse behavior via the MGBA. The findings underscore the significance of the gut microbiota, colon, and PFC, with PFC interneurons playing a crucial role in forced loss.

## Discussion

In this study, we combined a model of social dominance loss in mice with multi-omics analysis to investigate the influence of forced loss on the MGBA. We observed that forced loss mice exhibited alterations in microbial composition and function, primarily characterized by disruptions in *Muribaculaceae* and butanoate metabolism. Additionally, we found changes in signaling transmission within the colon, where these microbes reside. Notably, significant changes in single-nucleus transcription levels were primarily observed in interneurons, involving GABAergic synaptic transmission. By integrating 16S rRNA-seq, metagenomics, colon RNA-seq, and snRNA-seq data, we identified consistent alterations in intercellular signaling dysfunction across the gut microbiota, colon, PFC, and PFC interneurons. Our findings provide evidence for the MGBA perturbations induced by social status loss. Furthermore, these discoveries may offer insights into a better understanding of the pathophysiological mechanisms underlying brain disorders.

Social status has been shown to have a significant correlation with depression[50], and social status loss significantly elevates the probability of developing brain disorders in humans, such as depression[51]. Fan et al. utilized the tube test to create a paradigm of social status loss, thereby revealing the neural mechanisms linking social status loss to depression[27]. In our study, we found that forced loss could disrupt the abundance of *Muribaculaceae* and butanoate metabolism. The main metabolic capability of *Muribaculaceae* is the degradation of polysaccharides, which can produce SCFAs such as acetate and propionate[52-54]. Butyrate serves as an important medium for signal transduction and can influence the brain via the MGBA[13]. A study has demonstrated that sodium butyrate could alter the expression of brain-derived neurotrophic factor (BDNF) in mice, and prolonged treatment with sodium butyrate has been validated to induce antidepressant-like effects[55]. Butyrate could regulate orexin signaling in the lateral hypothalamus, playing a crucial role in sleep[56]. Prolonged low-dose lead exposure can lead to significant impairments in learning, memory, and cognitive functions, accompanied by a reduction in butyrate levels. Supplementation with butyrate can mitigate the learning and memory deficits caused by lead exposure[57]. Moreover, we observed that forced loss affected the regulation of the Notch signaling pathway in the intestine. The Notch signaling pathway is a critical signaling pathway for maintaining intestinal homeostasis[58], and Xue et al. found that the Notch signaling pathway could be modulated by probiotics to promote mucosal repair and improve intestinal homeostasis[59]. Therefore, the Notch signaling pathway may play a role in the MGBA's response to forced loss by influencing intestinal homeostasis.

Previous research has indicated that pyramidal neurons exert disinhibitory effects in social competition, influencing the social status of mice[23]. A study on the neurobiological mechanisms of social hierarchy has primarily focused on the regulation of neural circuits[28], highlighting the significance of neurons in social status modulation. However, the transcriptome changes at the single-nucleus level induced by social status remain largely unexplored. Here, we utilized snRNA-seq to identify specific transcriptional alterations in neurons and glial cells following forced loss. Interneurons were preferentially influenced in PFC. Consistent with our findings, Tan et al. revealed that silencing GABAergic interneurons in the dmPFC enhanced social status in mice[60], further emphasizing the significant role of interneurons in social status. Additionally, Huang et al. discovered the absence of gut microbiota led to significant changes in neuropsychiatric behaviors and neuroimmune dysfunction by causing alterations in the

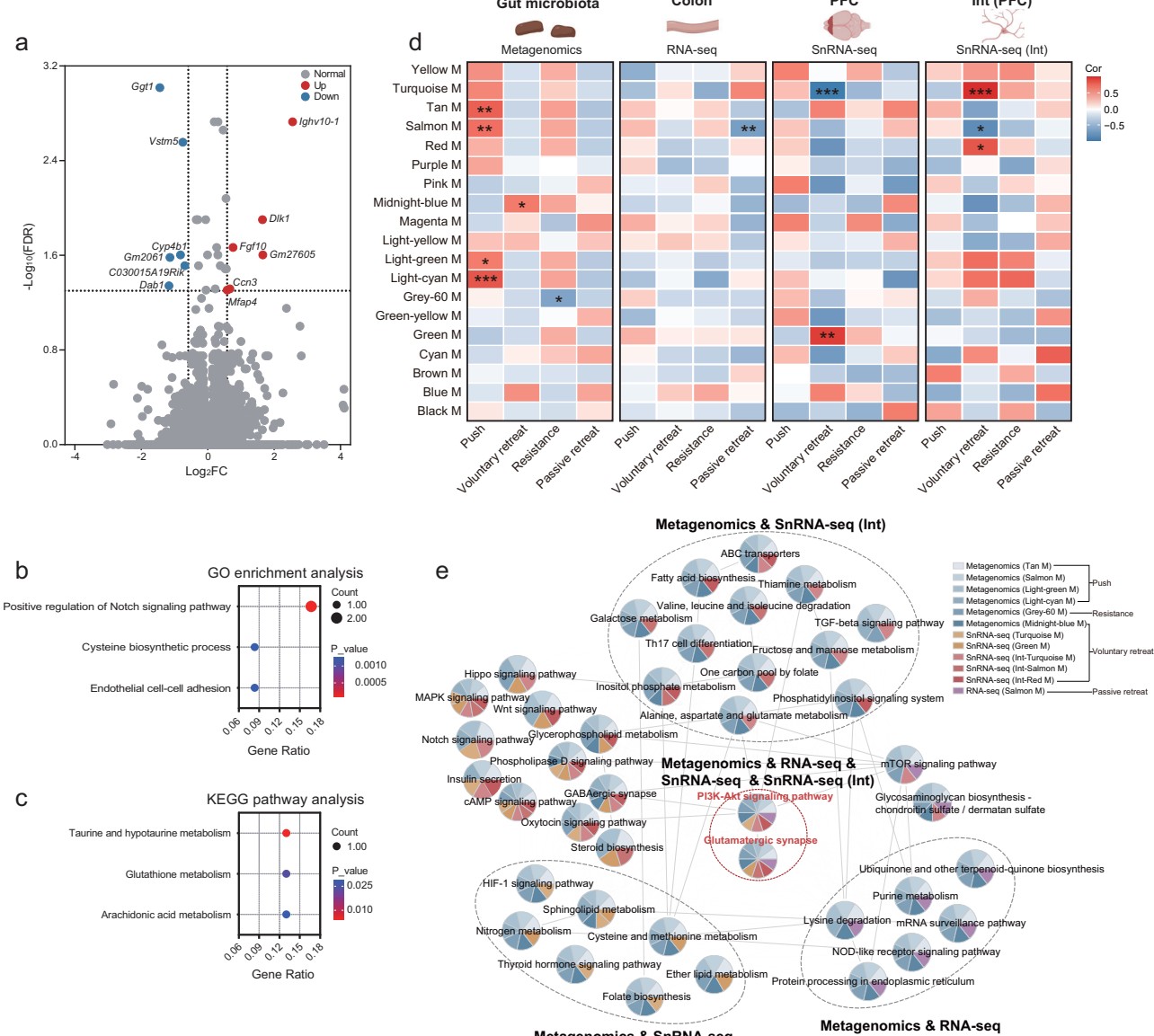

**Fig. 7 | Correlation between gut microbiota, colon, PFC, PFC interneurons, and behaviors related to forced loss. a** Volcano plot displayed DEGs between the go-through-tube group and the forced-loss group in RNA-seq (Go-through-tube, $n = 7$; Forced loss, $n = 7$; |log2(FC)| > 0.585, and FDR < 0.05). **b** The biological processes related to the DEGs in RNA-seq were identified by GO enrichment analysis ($p < 0.05$). **c** KEGG pathway analysis of DEGs in RNA-seq ($p < 0.05$). **d** Heatmap showed the correlations between WGCNA modules from the gut microbiota, colon,

PFC, PFC interneurons and behaviors related to the forced loss ($r > 0.6$ or $< -0.6$, *$p < 0.05$, **$p < 0.01$, ***$p < 0.001$, two-tailed Pearson correlation). The illustration was created in BioRender. Yang, R. (2025) https://BioRender.com/c35i595. **e** Co-expression network of KEGG pathways within WGCNA modules that were significantly associated with forced loss-related behaviors ($r > 0.6$ or $< -0.6$, and $p < 0.05$, two-tailed Pearson correlation).

transcriptome of microglial subpopulations[61]. A study has linked PFC synaptic transmission deficits to social impairments, emotional disturbances, and memory impairment in brain disorders such as autism, schizophrenia, depression, and AD[62]. By establishing a causal relationship between excitatory synaptic strength and social status, it was found that reducing AMPA receptor-mediated synaptic transmission could transform dominant mice into subordinate[63]. Our study suggested that forced loss impaired GABAergic synaptic transmission in PFC interneurons. Furthermore, our WGCNA of snRNA-seq data indicated that the phenotypes of forced loss-related behaviors involved diverse cell types and emphasized the role of neurons. Specifically, Interneurons showed a unique correlation with voluntary retreat behavior during the tube test, characterized by changes in synaptic transmission function and signaling pathways. Perturbations of signal transmission may provide insights into brain disorders induced by forced loss. In particular, *ErbB4* and *Grik1* were involved in

GABAergic synaptic transmission of DEG enrichment in interneurons. Consistent with our findings, *ErbB4* was expressed in the adult brain and maintained the excitation-inhibition balance by promoting or sustaining GABA release[64,65]. Moreover, *ErbB4* has been identified as a susceptibility gene for major depressive disorder, schizophrenia, and other disorders[66]. Additionally, selectively silencing *Grik1* within the adult amygdala has been shown to reduce GABAergic transmission and induce mild anxiety-like behavior[67].

Furthermore, our multi-omics analysis revealed that the PI3K-Akt signaling pathway might be an important pathway through which forced loss regulates behaviors in mice via the MGBA. Kanoski et al. reported that ghrelin signaling in the ventral hippocampus could enhance food intake in rats through the PI3K-Akt signaling pathway[68]. Additionally, the PI3K/Akt/GSK-3β signaling pathway may be the mechanism by which melatonin mitigates amyloid beta-induced memory deficits, tau

hyperphosphorylation, and neurodegeneration in mice[69]. Moreover, prebiotics can improve neuroinflammation, learning, and memory function induced by a high-fat diet through the IRS/PI3K/AKT signaling pathway[70]. Gu et al. demonstrated that *Prevotella copri* could activate the PI3K-Akt signaling pathway by increasing the levels of guanosine, a gut microbial metabolite, thereby promoting neurorehabilitation in mice with traumatic brain injury[71]. Therefore, the PI3K-Akt signaling pathway is involved in the regulation of the microbiota-gut-brain axis, and we consider it a potential pathway through which forced loss alters mice behaviors via MGBA.

We have discovered that forced loss can induce disruptions in the MGBA using a multi-omics research approach, providing evidence for further studies on how forced loss influences mental disorders. However, there are limitations to our findings. First, this study involved only male mice to circumvent the potential impact of estrogen. The impact of sex differences on social hierarchy behaviors and neural mechanisms still needs further exploration. Second, further quantification of gut microbiota-related metabolites, including butanoate, is needed to verify our findings. Third, several confounding factors could influence the structure of gut microbiota, such as the normalization of microbiota due to the co-housing effect and coprophagic behavior[72,73]. Whether social hierarchy is restored with the normalization of the gut microbiota remains to be further confirmed. Fourth, the gut microbiota itself possesses a complex compositional nature. We observed correlations among differential ASVs resulting from forced loss, and further studies, including in vitro co-culture experiments and functional metabolomics of microbial interactions, are required to elucidate the underlying interaction mechanisms. Finally, how the key microbiota mediated by forced loss acts on the CNS and interneurons via the MGBA needs further validation.

Taken together, our analysis of the gut microbiota composition and function revealed that forced-loss mice exhibit disturbances in *Muribaculaceae* and butanoate metabolism. By integrating metagenomics, colon RNA-seq, and snRNA-seq data, we identified consistent alterations in the PI3K-Akt signaling pathway and the glutamatergic synapse from the gut microbiota to the prefrontal cortex following forced loss. These findings provide evidence for understanding the mechanisms of the microbiota-gut-brain axis in neuropsychiatric disorders associated with forced loss. Furthermore, the snRNA-seq analysis highlighted the critical role of interneurons in behaviors related to forced loss. A focus on interneuron subtypes revealed that the Meis2, Meis2-Reln, and Pvalb subtypes might collectively influence forced loss. These findings provide a potential foundation for investigating the major neurological changes resulting from forced loss.

## Methods

### Ethics statement
This research protocol was reviewed and approved by the Ethics Committee of Chongqing Medical University (No. IACUC-CQMU-2024-0409). We have complied with all relevant ethical regulations for animal use. All experiments utilized 6-8-week-old male pathogen-free *C57BL/6J* mice sourced from Hunan SJA Laboratory Animal Co., Ltd. (Changsha, China). The sample size for each experiment was determined with reference to the literature[61,74]. A total of 46 mice were used. The number of mice was defined in the figure legends. All *C57BL/6J* male mice were bred and housed in the Experimental Animal Center of Chongqing Medical University. All mice were maintained under controlled conditions of temperature (22–24 °C) and humidity (50-60%), with a 12-hour light/dark cycle, and had free access to food and water.

### Antibiotic treatment
Antibiotic treatment was conducted following a customized protocol derived from established methodologies[75]. Daily intragastric administration to mice included neomycin sulfate at 200 mg/kg, ampicillin at 200 mg/kg, metronidazole at 200 mg/kg, and vancomycin at 100 mg/kg. To deplete the gut microbiota, antibiotic treatment lasted 7 consecutive days prior to forced loss or go-through-tube.

### Fecal DNA extraction and quantitative PCR (qPCR)
Fresh fecal pellets were aseptically collected to evaluate antibiotic-induced gut microbiota depletion. Total genomic DNA was extracted using the E.Z.N.A.® Stool DNA Kit (Omega Bio-tek, Norcross, GA, USA; Cat. *No.* D4015-01) following the manufacturer's protocol[76]. Quantitative PCR was performed using TB Green® Premix Ex Taq II[77] (Takara Bio, Shiga, Japan; Cat. No. RR820A) with universal bacterial 16S rRNA primers 27 F (5'-GAGAGTTTGATCCTGGCTCAG-3') and 1492 R (5'-TACGGC-TACCTTGTTACGAC-3')[78]. Each 20 μL reaction contained 10 μL 2× Premix, 0.8 μL each primer (10 μM), 2 μL fecal DNA (20 ng/μL), and 6.4 μL nuclease-free water. Thermal cycling comprised initial activation at 95 °C for 30 s (4.4 °C/s), 40 cycles of 95 °C for 5 s (4.4 °C/s) and 60 °C for 30 s (2.2 °C/s) with endpoint fluorescence acquisition, followed by melt curve analysis from 60 °C to 95 °C at 0.11 °C/s (continuous acquisition, 5 readings/°C), and final cooling to 50 °C for 30 s (2.2 °C/s).

### Tube test
As described previously[32], tube test is employed to assess the social status among mice (Fig. 1a). Prior to the tube test, the mice were trained to alternate through the tube 10 times over four days. This training phase is to acclimate the mice to the testing process and surroundings. During the testing phase, two mice from the same cage were placed at the opposing extremities of the tube, and were then permitted to move towards each other until they encountered in the middle. They were then simultaneously released, and the mouse compel its opponent to exit the tube was deemed the "winner", while the one that first retreat from the tube was considered the "loser". Four mice in the same cage were tested in pairs six times a day. Social status among the mice was defined by the number of wins, with rankings assigned as 1, 2, 3, and 4 accordingly. The behavior of the mice during each tube test was video-recorded for subsequent analysis.

### Forced loss
Forced loss was performed as previously described[27]. After identifying mice that consistently maintained rank 1 for at least four consecutive days, we randomly assigned these mice to either the forced-loss group or the go-through-tube group (Fig. 1a). Rank1 mice in the forced-loss group were positioned in the tube against subordinate individuals within the same cage (rank2, 3 and 4), with the exit of subordinate mice was blocked, thereby forcing rank1 mice to lose. This process was repeated ten times per day for four consecutive days. We attached a dynamometer to the blocker on the side of the subordinate mouse to measure the force and duration of the pushing applied to the subordinate mouse. We then performed the tube test to determine whether the original rank1 mice suffered a decline in their social hierarchy after the four-day forced-loss period. In contrast, rank1 mice in the go-through-tube group alternated through the tube ten times each day for four days without experiencing forced loss. The process of forced loss and the final tube test on the fifth day were video-recorded for subsequent analysis.

### Analysis of tube test behavior
A camera was positioned 30 cm away from the tube to record a lateral view of the entire tube, recording the behavior of the mice during the tube test. To minimize potential confounding factors in tests, we conducted each test at the same time and used the same equipment. We analyzed the video recordings of the rank1 mouse before and after forced loss or go-through-tube. The duration and proportion of "push", "voluntary retreat", "resistance", "passive retreat" and "stillness" behaviors of rank1 mice were recorded and analyzed statistically. "Push" refers to one mouse extending its head noticeably forward to push against the opponent or inserting its head beneath the opponent to push forward. "Voluntary retreat" was defined as the rank1 mouse voluntarily retreating when it is not being pushed by the opponent. "Resistance" was defined as one mouse maintaining territory without retreating when pushed by its opponent, usually with its head being elevated higher by an opponent. "Passive retreat" was described as the rank1

mouse retreating when pushed by the opponent. "Stillness" was characterized by neither mouse doing anything other than sniffing.

### Open-field test (OFT)

Each mouse was separately placed inside an open cube ($42 \times 42 \times 42$ cm) with a white background[61]. A six-minute video was recorded for subsequent analysis. The total distance each mouse moved was utilized to assess the motor ability. Data analysis was performed using Noldus software.

### Sample preparation

After deep anesthesia and euthanasia, the mice's brains were rapidly removed. Subsequently, the prefrontal cortex was isolated and immediately immersed in liquid nitrogen to facilitate rapid freezing, ensuring the nuclei remained intact. The samples were then stored at $-80\ °C$ for long-term preservation. The cryopreserved tissue samples were rinsed with ice-cold PBSE (PBS supplemented with 2 mM EGTA). The nuclei were isolated according to the instructions provided by Singleron Biotechnologies using GEXSCOPE® Nuclei Isolation Reagent (Nanjing, China). After isolation, the nuclei were resuspended in PBSE at a density of one million per 400 μL. This suspension was then passed through a 40 μm cell strainer for purification, and the nuclei concentration was determined by trypan blue exclusion before staining with DAPI (1:1000) from Thermo Fisher Scientific (Cat. No. D1306)[79].

The upper segments of the colon were excised from the mice after anesthesia, and the colon contents were immediately removed and dispensed into sterile tubes. Both the colon contents and the colonic tissue from the upper segment were rapidly immersed in liquid nitrogen for freezing and then stored at $-80\ °C$.

### 16S rRNA gene sequence analysis

The procedure for preparing and sequencing the 16S rRNA gene amplicon library was carried out[80]. Raw sequencing reads were subjected to quality control via fastq software, followed by merging with FLASH software. Based on the recommended parameters, noise reduction was applied to the optimized sequences post-quality control and assembly using the DADA2 plugin within the Qiime2 pipeline. Amplicon Sequence Variant (ASV) is a unit used to represent microbial species or populations. Utilizing DADA2 denoising processing, DNA sequences devoid of chimeras and sequencing errors were obtained. DNA sequences within microbiomes are measured and analyzed to calculate their differences and similarities. Similar DNA sequences are clustered into the same ASVs, thereby obtaining an ASV number for each microorganism. Utilizing the Silva 16S rRNA gene database (version 138), taxonomic analysis of ASVs was performed using the Naïve Bayes classifier available in Qiime2.

Differences in ASV abundance across the two groups were analyzed by the Wilcoxon rank-sum test. The Ace index for species richness and Shannon and Simpson indices for species diversity were used to measure microbial community diversity. The analysis of group differences in Alpha diversity was conducted with the Wilcoxon rank-sum test. To examine the similarity of microbial community structures among samples, principal coordinates analysis (PCoA) based on the abund_jaccard distance metric was employed. Furthermore, PERMANOVA nonparametric testing was combined with the PCoA analysis to evaluate the statistical significance of distinctions in microbial community structures among sample groups.

### Metagenomic analysis

The metagenomic analysis began with the extraction of DNA, then proceeded to the establishment of paired-end (PE) library, bridge PCR, and sequencing[81]. Subsequently, quality trimming was performed using fastp to retain high-quality paired-end reads and single-end reads. Using the BWA software (http://bio-bwa.sourceforge.net, version 0.7.17), the reads were aligned to the host DNA sequences, and reads with high alignment similarity to contaminants were excluded. The refined sequences were then assembled with MEGAHIT (https://github.com/voutcn/megahit, version 1.1.2), selecting contigs of at least 300 bp as the final assembly output. In these configurations, the open reading frames were identified by Prodigal

(https://github.com/hyattpd/Prodigal, version 2.6.3). Genes exceeding or equal to 100 bp in nucleotide length were selected and translated into their corresponding amino acid sequences.

Using CD-HIT (http://weizhongli-lab.org/cd-hit/, version 4.7), these predicted gene sequences were then clustered to establish a non-redundant gene catalog. SOAPaligner (https://github.com/ShujiaHuang/SOAPaligner, version soap2.21) was used to calculate gene abundance. Finally, the amino acid sequences from the non-redundant gene catalog were aligned against the KEGG database, and the abundance of corresponding functions was quantified by calculating the gene abundance associated with KEGG Orthology, pathways, enzyme commission numbers, and modules.

### Transcriptome sequencing analysis

Total RNA extraction was performed using Trizol reagent (Invitrogen), and following the manufacturer's protocol, the cDNA library was prepared utilizing the VAHTS Universal V6 RNA-seq Library Prep Kit from Illumina (vazyme, Inc.). The library underwent quality control using the Agilent 2200 platform, followed by paired-end sequencing at 150 bp using the DNBSEQ-T7 sequencer. During the RNA-Seq data analysis process[82–84], we used Hisat2 as the RNA alignment algorithm to map the filtered sequencing reads to the reference genome (mm10_Ensembl) corresponding to the sequenced species (Taxonomy ID: 10090) to determine their genomic locations. RUVSeq was utilized to remove unwanted variation from the RNA-Seq data. The DESeq2 algorithm was utilized to identify DEGs with the criteria of $|\log 2(FC)| > 0.585$; FDR < 0.05. GO enrichment analysis was typically conducted using Fisher's exact test and chi-squared ($\chi^2$) test for classification. The $p$-values obtained from these tests were corrected for false discovery rate (FDR), and lower FDR suggests a smaller error in the assessment of the $p$-values. Significant pathways for the DEGs were identified based on the KEGG database. Pathway annotations for microarray genes were downloaded from the KEGG database (http://www.genome.jp/Kegg/). To identify pathways with significant enrichment, Fisher's exact test was applied, and the resulting $p$-values were subsequently adjusted by the Benjamini-Hochberg (BH) FDR algorithm. For results to be considered statistically significant, the adjusted $p$-values needed to be less than 0.05.

### Preparation of single-nucleus RNA sequencing library

Using a microfluidic chip (GEXSCOPE® Single Nucleus RNA-seq Kit, Singleron Biotechnologies)[85], a mononuclear suspension with a concentration of $3–4 \times 10^5$ nuclei/mL was processed, and snRNA-seq libraries were then generated according to the manufacturer's instructions. Sequencing was conducted on an Illumina NovaSeq 6000 system, configured for a read length of 150 base pairs at the end of each fragment.

### Single-nucleus RNA sequencing and analysis

The raw reads were processed using the default parameters of CeleScope (Singleron Biotechnologies, version 1.5.2) to generate gene expression profiles[86]. From R1 reads, barcodes and unique molecular identifiers (UMIs) were extracted and corrected. The adapter sequences and poly-A tails were clipped from R2 reads, which were then aligned were aligned to the GRCh38 (hg38) or GRCm38 (mm10) transcriptome using STAR (version 2.6.1b). Subsequently, FeatureCounts (version 2.0.1) was used to assign uniquely mapped reads to genes. Successfully assigned reads with identical cell barcodes, UMIs, and genes were combined to compile the gene expression matrix for subsequent analysis.

Using the Seurat package (version 4.3.3), scaled data was obtained after normalization and regression at the cell level. A total of 29 unsupervised cell clusters were identified through graph-based clustering methods. DEGs were calculated using the FindAllMarkers (Wilcoxon) method according to default criteria, and those marker genes satisfying $|\log_2(FC)| > 0.25$, and FDR < 0.05 (Bonferroni correction) were selected. The 29 clusters were further annotated into several major cell types based on their marker genes. Gene function enrichments were explored through GO enrichment analysis (geneontology.org). Functions fulfilling FDR < 0.05 were presented as significant entries.

## Cell communication analysis

The intercellular interactions in the PFC were analyzed using the CellChat package (version 2.1.2). The CellChat package includes a comprehensive ligand-receptor interaction database. Through the integration of gene expression data with the existing database, which comprised known interactions of signaling ligands, receptors, and their cofactors, we modeled the probability of intercellular communication[40] and identified statistically significant ($p < 0.05$) intercellular communications.

## Weighted gene co-expression network analysis

Using the WGCNA package (version 1.72), we aimed to analyze gene modules associated with behaviors in the gut microbiota, colon, and PFC[87]. This process involved importing both gene expression data and trait data, assessing data quality and detecting outliers, and selecting an appropriate soft thresholding parameter ($\beta$) value. Typically, the minimum $\beta$ value satisfying $R^2 > 0.9$ or closest to 0.9 was chosen ($\beta$ value: 12, gut microbiota; 16, colon; 30, PFC; 13, PFC interneurons; 11, PFC excitatory neurons; 13, PFC microglia; 21, PFC astrocytes; 20, PFC oligodendrocytes; 8, PFC oligodendrocyte precursors). To minimize noise and false correlations, the gene expression matrix was converted into a topological overlap matrix (TOM). The resulting TOM matrix represented the weighted correlation coefficients between genes. The weighted gene matrix was then clustered, and the gene count for each module was defined. Different modules were clustered, and a module-trait correlation plot was generated. The obtained modules demonstrated an association with behaviors related to forced loss, and modules that showed a significant correlation ($r > 0.6$ or $< -0.6$ and $p < 0.05$) were considered behavior-related modules.

## Statistics and reproducibility

The statistical data were all analyzed using GraphPad Prism software (GraphPad Software, La Jolla, CA; version 9.5.0) and R studio (version 4.3.3). A two-tailed unpaired Mann–Whitney test was used to compare two groups in terms of rank changes, behaviors, force and duration, proportion of cell type, and DEGs downsampling analysis. An unpaired $t$-test was employed to compare the two groups in terms of body weight, total distance, and relative DNA expression levels. To compare four groups, Kruskal–Wallis test (Tukey post hoc test) was utilized. The Chi-square test was used for the success rates of demoting dominant mice to subordinate status and retaining rank1 between the two groups. A $p$-value < 0.05 was considered statistically significant. We did not exclude any data points because there were no extreme values during the data analysis process. The experimenter was blinded to the allocation information until the completion of statistical analyses. Additional information regarding sample size and raw data is provided in the supplementary data.

## Reporting summary

Further information on research design is available in the Nature Portfolio Reporting Summary linked to this article.

## Data availability

All data supporting the findings of this study are available within the paper and its supplementary information (Supplementary Data 1–2). The data that support the findings of this study are available from the corresponding author upon reasonable request. The 16S rRNA sequencing data generated by this study is available in the Gene Expression Omnibus (https://www.ncbi.nlm.nih.gov/geo/query/acc.cgi?acc=GSE289698). The raw data of metagenomics and single-nucleus RNA sequencing are available in the Sequence Read Archive (https://www.ncbi.nlm.nih.gov/bioproject/PRJNA1223975, https://www.ncbi.nlm.nih.gov/bioproject/PRJNA1226127).

## Code availability

All data analysis procedures were described in the methods. The code for analysis can be found at https://doi.org/10.5281/zenodo.14927861[88].

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

## Acknowledgements
This work was supported by the National Key R&D Program of China (STI2030-Major Projects 2021ZD0202400, STI2030-Major Projects 2021ZD0200600), the National Natural Science Foundation Project of China (82201688, 82171523, 82471545, 82401784, 32400850, 82401523), National Reserve Talent Project in the Health and Wellness Sector of Chongqing (HBRC202410, HBRC202417), China Postdoctoral Science Foundation (2024MD754023), the Program for Youth Innovation in Future Medicine of Chongqing Medical University, Science and Technology Research Program of Chongqing Municipal Education Commission (Grant No. KJZD-K202400404), the Key Project of the Natural Science Foundation of Chongqing (Chongqing Science and Technology Development Foundation) under Grant No. 2024NSCQ-KJFZZDX0005, the New Chongqing Youth Innovation Talent Project (Life and Health) under Grant No. 2024NSCQ-qncxX0029, Joint Project of Chongqing Health Commission and Science and Technology Bureau (2024QNXM046).

## Author contributions
Designed the experiments: J.W. and P.Z. Operated experiments: R.J.Y., X.W. Performed the 16S rRNA gene sequence and metagenomic analysis: R.J.Y., X.W., Y.F.L., X.M.T. Performed the transcriptome sequencing analysis: R.J.Y., X.W., M.H.Y. Performed the snRNA-seq analysis: R.J.Y., X.W., J.Y., X.Y.Z., Y.H., J.P.Z., P.L. Animal behaviors analysis: R.J.Y., X.W., Y.Y.W., X.Y.Z. Drafted the paper: R.J.Y., X.W. Revised the paper for intellectual content: J.W. and P.Z.

## Competing interests
The authors declare no competing interests.
