## [Transparent Peer Review file · Communications Biology]

Perturbations in the microbiota-gut-brain axis shaped by social status loss

Corresponding Author: Professor Jing Wu

Version 0:

Reviewer comments:

Reviewer #1

(Remarks to the Author)

Wu and colleagues provide a very elegant manuscript, well presented with clear figures describing the impact of social loss on the brain, the gut microbiome using multiple techniques.

Overall the paper is well presented, but a few clarifications might help the manuscript.

N numbers are not in every figure legend, it seems unclear in some experiments how many animals were used.

Did the authors consider quantifying butanoate?

Was any consideration given to using antibiotics in the study to confirm if depletion of Muribaculaceae had any impact on the outputs tested.

The titles of the figures should be more descriptive.

Why were only male mice used?

were animals kept in the same cages throughout the behaviour paradigms, do the authors think that shared environments should have normalised the impact of the microbiome given there would have been microbial transfer?

Line 32, introduction spelt incorrectly

Line 87 – cage

Line 131, please define what ASVs are?

Line 172, grammar

Reviewer #2

(Remarks to the Author)

This manuscript explores the relationship between social status loss and the gut microbiota-gut-brain axis, presenting a multi-omics approach to investigate how stress-induced changes in the microbiota may influence brain function. The major claims, particularly the involvement of the gut microbiota in mediating stress responses caused by social status loss, are intriguing and align with emerging interests in the gut-brain axis. This topic will likely appeal to the journal's readership and the broader community of researchers studying microbiota, stress, and brain health.

However, while the work is novel and has potential to influence thinking in the field, several aspects require further evidence to convincingly support the authors' conclusions:

The paper presents novel findings, particularly the correlations between gut microbiota and changes in prefrontal cortex interneurons. These insights are promising and could lead to future therapeutic strategies. However, the study's observational nature limits its ability to establish causality, particularly regarding the active role of the microbiota in the

phenotype. Experimental validation is necessary to strengthen the conclusions. For example, gut microbiota perturbation through antibiotic treatment, germ-free or gnotobiotic models, or fecal microbiota transplantation would substantiate the authors' hypothesis.

The lack of direct experimental evidence weakens the claim that the microbiota actively drives stress responses. The authors should consider testing their hypothesis experimentally to establish causality (e.g., fecal transplants or targeted colonization with identified species). Addressing whether the observed changes in the microbiota are a cause or a consequence of stress. Moreover, methodological details are occasionally unclear, such as how significant ASVs were identified. These aspects should be clarified in the text.

Additional Concerns and Suggestions

Figure 1: The bar plot representing mean \pm SEM is not appropriate given that the authors used a non-parametric test due to the non-normal distribution of the data. I recommend replacing it with boxplots, which are better suited for such data.

Line 122: The lack of differences in microbial community structure (Fig. S1) may partially be attributed to the coprophagic behavior of mice. This should be acknowledged here or discussed later in the manuscript.

Figure 2C: The heatmap does not show the relative abundance of genera as stated in the figure legend but instead displays fold changes and correlations between genera and behaviors. Additionally, it is unclear whether the x-axis represents multiple ASVs per genus. If so, these should be reported together with their assigned taxa for clarity.

The method used to identify significantly different ASVs is unclear. Which test or tool was employed? This should be explicitly stated in the results, figure legends, and methods section.

Figure 2D: The correlations among ASVs in the two clusters could result from the compositional nature of microbiome data. This limitation should be acknowledged here and in the discussion. Any conclusions drawn from these correlations should ideally be tested experimentally.

Figure 2E: The LEfSe analysis appears forced to yield significant results. By default, LEfSe uses an LDA score of 2 as a threshold. Additionally, LEfSe applies a Wilcoxon test and reports nominal p-values without accounting for multiple comparisons. I suggest first testing for significant differences using a Wilcoxon rank-sum test corrected for multiple comparisons and then applying LEfSe on the significantly different pathways to identify the most biologically relevant ones, aligned with the hypothesis.

Version 1:

Reviewer comments:

Reviewer #1

(Remarks to the Author)

The authors have provided a very comprehensive rebuttal to my comments, I am happy to approve this paper for publication.

Reviewer #2

(Remarks to the Author)

I thank the authors for addressing my comments

Thank you very much for the feedback and comments on our manuscript titled "Perturbations in the microbiota-gut-brain axis shaped by social status loss". Those comments are all valuable and very helpful for revising and improving our paper. We have carefully considered all the suggestions and comments. Based on these valuable comments, we have solidified our arguments, such as administering antibiotics via gavage to deplete gut microbiota, thereby demonstrating that alterations in the gut microbiota are a cause of the stress responses. The revised sections in the manuscript and figure legends are highlighted in red font.

Responds to the reviewers' comments:

Reviewer #1

***General Assessment:** Wu and colleagues provide a very elegant manuscript, well presented with clear figures describing the impact of social loss on the brain, the gut microbiome using multiple techniques.*

Response: Thank you for your positive comments. We found that stress altered the composition and function of the gut microbiota, increasing *Muribaculaceae* abundance and enhancing butanoate metabolism. We further validated the crucial role of gut microbiota in forced loss by depleting the gut microbiota with antibiotics. We have answered all concerns point-by-point, and revised the manuscript accordingly.

***Comment 1.1:** N numbers are not in every figure legend, it seems unclear in some experiments how many animals were used.*

Response 1.1: We appreciate this reviewer's careful reading of our manuscript. We have modified the legends for Fig. 1f, Fig. 2a, Fig. 2e, Fig. 4d, Fig. 6e, Fig. 7a, Fig. S1a, Fig. S1b, Fig. S1d, Fig. S1e, Fig. S2b, and Fig. S3a by adding the number of experimental animals in each group (line 830; 836-837; 847-848; 881; 911; 921; 934-935; 937-938; 943; 945-946; 961; 973).

***Comment 1.2:** Did the authors consider quantifying butanoate?*

Response 1.2: We sincerely thank the reviewer for careful reading and question. In this study, we found that forced loss altered the composition and function of the gut microbiota, and gut microbial depletion resisted forced loss-induced hierarchical demotion and behavioral alteration. We agree that quantification of butanoate is useful. Wang et al. found that patients with insomnia exhibit lower serum butyrate levels. Butyrate could regulate orexin signaling in the lateral hypothalamus, playing a crucial role in sleep¹. Prolonged low-dose lead exposure can lead to significant impairments in learning, memory, and cognitive functions, accompanied by a reduction in butyrate levels. Supplementation with butyrate can mitigate the learning and memory deficits caused by lead exposure². We consider targeted metabolomics in future studies, specifically by quantifying butanoate to further validate our findings and explore its relationship with the microbiome and behaviors. We have explained this part in the discussion (line 343-347; 395-396).

***Comment 1.3:** Was any consideration given to using antibiotics in the study to confirm if depletion of *Muribaculaceae* had any impact on the outputs tested.*

Response 1.3: Thank you for your constructive suggestions for us to improve the manuscript. To determine the critical role of gut microbiota in forced loss-induced social hierarchy and behavioral alteration, we further conducted supplementary experiments to demonstrate that antibiotic-mediated depletion of the gut microbiota can influence the outcomes of forced loss.

We compared microbiota-intact and microbiota-depleted mice, and the microbiota-depleted mice were generated through intragastric administration of broad-spectrum antibiotics (ABX). The composition of the antibiotic cocktail includes vancomycin (100 mg/kg), neomycin sulphate (200 mg/kg), metronidazole (200 mg/kg), and ampicillin (200 mg/kg)³. Quantitative PCR results following antibiotic gavage indicated that the gut microbiota was effectively depleted (Response Fig. 1e). Subsequently, these mice underwent the forced loss paradigm. Compared to microbiota-intact mice, microbiota-depleted mice exhibited marked resistance to hierarchical demotion after forced loss (Response Fig. 1b). While 76.92% of dominant (rank1) microbiota-intact mice were demoted to subordinate status following forced loss, only 28.57% of microbiota-depleted mice lost dominance status (chi-square test, $\chi^2 = 2.105$, $p = 0.0353$) (Response Fig. 1c top). After forced loss, the proportion of rank1 mice retaining dominance increased from 7.69% in microbiota-intact group to 42.86% in microbiota-depleted group (Response Fig. 1c, bottom). Dominant mice with depleted microbiota gradually exerted less force against subordinate mice over time, while the duration of the resistance remained unchanged (Response Fig. 1d). Notably, during the day 2 to 4 of the forced loss process, microbiota-depleted dominant mice displayed elevated force and prolonged duration toward subordinates compared to microbiota-intact dominant mice (Response Fig. 1g). Besides, microbiota-depleted mice predominantly exhibited pushing rather than retreating, and microbiota-depleted mice exhibited stable behaviors before and after forced loss, with no significant difference in pushing, voluntary retreating, or resistance behaviors (Response Fig. 1f). Importantly, compared to microbiota-intact mice, microbiota-depleted mice exhibited increased pushing, reduced voluntary and passive retreating after forced loss (Response Fig. 1h). Collectively, these findings indicate that gut microbiota depletion allows mice to retain dominance, resisting both hierarchical demotion and behavioral alteration induced by forced loss (line 159-186; 426-443).

Response Figure 1

Fig. R1 Gut microbiota depletion regulated hierarchical and behavioral alteration induced by forced loss. **a**, Schematic of the experimental timeline: Antibiotic-mediated gut microbiota depletion (ABX) followed by forced loss intervention (Forced loss + ABX, $n = 7$; Go-through-tube + ABX, $n = 6$). The illustrations were created with BioRender (<https://biorender.com>). **b**, Daily tube test outcomes in microbiota-depleted mice before and after forced loss (Mann-Whitney test, two-tailed). **c**, Success rates of demoting dominant mice to subordinate status (top) and retaining rank1 (bottom) in microbiota depleted versus microbiota-intact mice after forced loss (Forced loss, $n = 13$; Forced loss + ABX, $n = 7$). Statistical significance determined by chi-square test (Success rate of forced loss: $\chi^2 = 2.105$, $p = 0.0353$; Rank retention rate: $\chi^2 = 3.516$, $p = 0.0608$). **d**, Average force (N) and duration (s) per trial during forced loss in microbiota-depleted mice ($n = 7$ mice, 21 trials, Mann-Whitney test, two-tailed). **e**, DNA expression of gut microbiota genes in the control and ABX groups (Con, $n = 8$; ABX, $n = 14$; Con vs. ABX, $****p < 0.0001$; data are mean \pm SEM; T test, two-tailed). **f**, Time percentage of pushing, voluntary retreating, resistance (when pushed by an opponent), and passive retreating (when pushed by an opponent) during tube test for microbiota-depleted mice before and after go-through-tube or forced loss procedure (Go-through-tube + ABX, $n = 6$ mice, 18 trials; forced loss + ABX, $n = 7$ mice, 21 trials). $*p < 0.05$, $**p < 0.01$, $***p < 0.001$,

**** $p < 0.0001$, ns, not significant; data are mean \pm SEM; Mann-Whitney test, two-tailed). **g**, Comparison of average force (left) and duration (right) per trial during forced loss between microbiota-depleted and microbiota-intact mice (Forced loss, $n = 13$ mice, 10 trials / mice; Forced loss + ABX, $n = 7$ mice, 10 trials / mice; Mann-Whitney test, two-tailed). **h**, Time percentage for pushing, voluntary retreating, resistance (when pushed by an opponent), and passive retreating (when pushed by an opponent) during tube tests in microbiota-depleted versus microbiota-intact mice before and after forced loss (Forced loss, $n = 13$ mice, 39 trials; Forced loss + ABX, $n = 7$ mice, 21 trials; Mann-Whitney test, two-tailed).

Data are presented as mean \pm SEM. * $p < 0.05$, ** $p < 0.01$, *** $p < 0.001$, **** $p < 0.0001$; ns, not significant.

Comment 1.4: The titles of the figures should be more descriptive.

Response 1.4: Following this reviewer's suggestion, we have already updated the figures for Fig. 4, Fig. 5 and Fig. S2 to provide more detailed descriptions, ensuring that editor, reviewers, and readers can understand the corresponding figures more clearly (line 875; 883; 958).

Comment 1.5: Why were only male mice used?

Response 1.5: Thanks for your question. Female mice are more susceptible to hormonal influences compared to male mice. Newhouse and Albert et al. revealed that group-housed female mice exhibit longer estrous cycles, with significant fluctuations in estrogen and progesterone levels. These fluctuations can affect experimental outcomes and increase data variability⁴⁻⁶. Therefore, we chose male mice to establish our model to avoid the influence of estrogen. Additionally, most studies use male mice to investigate the mechanisms underlying the development and progression of social hierarchy changes. By using adult male C57BL/6J mice, Fan et al. investigated the core neural circuit mechanisms underlying social status loss and the onset of depressive behaviors due to forced loss⁷. Choi et al. used adult male mice to study the molecular mechanisms by which different neuronal subpopulations in the medial prefrontal cortex encode various social competitive behaviors⁸. A study explored how astrocyte-neuron communication is involved in the establishment of social status in adult male mice⁹. We have explained this part in the discussion, and the impact of sex differences on social hierarchy behaviors and neural mechanisms still needs further exploration (line 393-395).

Comment 1.6: were animals kept in the same cages throughout the behavior paradigms, do the authors think that shared environments should have normalized the impact of the microbiome given there would have been microbial transfer?

Response 1.6: We thank the reviewers for their valuable questions. The evaluation of rank is limited to within the same cage ($n = 4$ mice per cage), and the process of forced loss is also conducted within the same cage. We agree with the reviewer's opinion that the co-housing effect can lead to the transfer of microbiota among mice, which normalize their microbiota. Studies have shown that the gut microbial diversity was reduced in non-coprothagic mice, while coprophagy normalized the microbiota of co-housed mice^{10,11}. Mice housed together for long periods will establish a social hierarchy, which is an inherent characteristic of co-housed groups¹². The tube test is a quantitative method for assessing

the social status of mice. We found that there were no significant differences in the gut microbiota among co-housed mice of different ranks (Response Fig. 2a, b). Therefore, co-housing induced gut microbiota normalization does not affect social status without intervention. In this study, we found that stress altered the composition and function of the gut microbiota, increasing *Muribaculaceae* abundance and enhancing butanoate metabolism, and gut microbial depletion resisted forced loss-induced hierarchical demotion and behavioral alteration. Whether social hierarchy is restored with the normalization of the gut microbiota remains to be further confirmed. We have addressed this part in the discussion (line 396-399).

Reponse Figure 2

Fig. R2 Rank evaluation and forced loss in a cage. a, Schematic diagram showed the process of obtaining mice of various ranks via the tube test, establishing forced loss and control groups among rank1 mice through forced loss or go-through-tube procedure, followed by sample sequencing. The illustration was created with BioRender (<https://biorender.com>). b, Daily tube test results for mice with relatively stable ranks.

Comment 1.7: Line 32, introduction spelt incorrectly

Response 1.7: We apologize for this error. In our resubmitted manuscript, the typo is revised (line 33).

Comment 1.8: Line 87 - cage

Response 1.8: We corrected this typo (line 89).

Comment 1.9: Line 131, please define what ASVs are?

Response 1.9: Amplicon Sequence Variant (ASV) is a unit used to represent microbial species or populations. Utilizing DADA2 denoising processing, DNA sequences devoid of chimeras and sequencing errors were obtained. DNA sequences within microbiome are measured and analyzed to calculate their differences and similarities. Similar DNA sequences are clustered into the same ASVs, thereby obtaining an ASV number for each microorganism. We have revised this part in the 16S rRNA gene sequence analysis method (line 508-512).

Comment 1.10: Line 172, grammar

Response 1.10: The text has been revised to: "Compared with the go-through-tube group, there was no proportional difference in main cell types in the forced loss group." (line 199-200).

Reviewer #2

General Assessment: This manuscript explores the relationship between social status loss and the gut microbiota-gut-brain axis, presenting a multi-omics approach to investigate how stress-induced changes in the microbiota may influence brain function. The major claims, particularly the involvement of the gut microbiota in mediating stress responses caused by social status loss, are intriguing and align with emerging interests in the gut-brain axis. This topic will likely appeal to the journal's readership and the broader community of researchers studying microbiota, stress, and brain health.

Response: We appreciate this reviewer's support for our research. We found that forced loss altered the composition and function of the gut microbiota, further confirming that changes in the microbiota contribute to the stress response. Single-nucleus transcriptomic analysis of the prefrontal cortex revealed that social status loss primarily affected interneurons, altering GABAergic synaptic transmission. Weighted Gene Co-expression Network Analysis indicated consistent changes in the PI3K-Akt signaling pathway and the glutamatergic synapse of the microbiota-gut-brain axis. We have answered all concerns point-by-point, provided additional experimental data, and revised the manuscript accordingly.

Comment 2.1: The paper presents novel findings, particularly the correlations between gut microbiota and changes in prefrontal cortex interneurons. These insights are promising and could lead to future therapeutic strategies. However, the study's observational nature limits its ability to establish causality, particularly regarding the active role of the microbiota in the phenotype. Experimental validation is necessary to strengthen the conclusions. For example, gut microbiota perturbation through antibiotic treatment, germ-free or gnotobiotic models, or fecal microbiota transplantation would substantiate the authors' hypothesis.

The lack of direct experimental evidence weakens the claim that the microbiota actively drives stress responses. The authors should consider testing their hypothesis experimentally to establish causality (e.g., fecal transplants or targeted colonization with identified species). Addressing whether the observed changes in the microbiota are a cause or a consequence of stress.

Response 2.1: Thank you for your constructive questions and comments. To determine the critical role of the gut microbiota in the social hierarchy and behavioral changes caused by forced loss, we validated the forced loss paradigm to establish the causal relationship between the gut microbiota and stress response.

We compared microbiota-intact and microbiota-depleted mice, and the microbiota-depleted mice were generated through intragastric ABX (antibiotic cocktail)³. Quantitative PCR results following antibiotic gavage indicated that the gut microbiota was effectively depleted (Response Fig. 1e). Subsequently, these mice underwent the forced loss paradigm. Compared to microbiota-intact mice, microbiota-depleted mice exhibited marked resistance to hierarchical demotion after forced loss (Response Fig. 1b). While

76.92% of dominant (rank1) microbiota-intact mice were demoted to subordinate status following forced loss, only 28.57% of microbiota-depleted mice lost dominance status (chi-square test, $\chi^2 = 2.105$, $p = 0.0353$) (Response Fig. 1c top). After forced loss, the proportion of rank1 mice retaining dominance increased from 7.69% in microbiota-intact group to 42.86% in microbiota-depleted group (Response Fig. 1c, bottom). Dominant mice with depleted microbiota gradually exerted less force against subordinate mice over time, while the duration of the resistance remained unchanged (Response Fig. 1d). Notably, during the day 2 to 4 of the forced loss process, microbiota-depleted dominant mice displayed elevated force and prolonged duration toward subordinates compared to microbiota-intact dominant mice (Response Fig. 1g). Besides, microbiota-depleted mice predominantly exhibited pushing rather than retreating, and microbiota-depleted mice exhibited stable behaviors before and after forced loss, with no significant difference in pushing, voluntary retreating, or resistance behaviors (Response Fig. 1f). Importantly, compared to microbiota-intact mice, microbiota-depleted mice exhibited increased pushing, reduced voluntary and passive retreating after forced loss (Response Fig. 1h). These findings further demonstrated that changes in the gut microbiota are a cause of stress responses (line 159-186; 426-443).

Response Figure 1

Fig. R1 Gut microbiota depletion regulated hierarchical and behavioral alteration induced by forced loss. **a**, Schematic of the experimental timeline: Antibiotic-mediated gut microbiota depletion (ABX) followed by forced loss intervention (Forced loss + ABX, $n = 7$; Go-through-tube + ABX, $n = 6$). The illustrations were created with BioRender (<https://biorender.com>). **b**, Daily tube test outcomes in microbiota-depleted mice before and after forced loss (Mann-Whitney test, two-tailed). **c**, Success rates of demoting dominant mice to subordinate status (top) and retaining rank1 (bottom) in microbiota depleted versus microbiota-intact mice after forced loss (Forced loss, $n = 13$; Forced loss + ABX, $n = 7$). Statistical significance determined by chi-square test (Success rate of forced loss: $\chi^2 = 2.105$, $p = 0.0353$; Rank retention rate: $\chi^2 = 3.516$, $p = 0.0608$). **d**, Average force (N) and duration (s) per trial during forced loss in microbiota-depleted mice ($n = 7$ mice, 21 trials, Mann-Whitney test, two-tailed). **e**, DNA expression of gut microbiota genes in the control and ABX groups (Con, $n = 8$; ABX, $n = 14$; Con vs. ABX, **** $p < 0.0001$; data are mean \pm SEM; T test, two-tailed). **f**, Time percentage of pushing, voluntary retreating, resistance (when pushed by an opponent), and passive retreating (when pushed by an opponent) during tube test for microbiota-depleted mice before and after go-through-tube or forced loss procedure (Go-through-tube + ABX, $n = 6$ mice, 18 trials; forced loss + ABX, $n = 7$ mice, 21 trials. * $p < 0.05$, ** $p < 0.01$, *** $p < 0.001$, **** $p < 0.0001$, ns, not significant; data are mean \pm SEM; Mann-Whitney test, two-tailed). **g**, Comparison of average force (left) and duration (right) per trial during forced loss between microbiota-depleted and microbiota-intact mice (Forced loss, $n = 13$ mice, 10 trials / mice; Forced loss + ABX, $n = 7$ mice, 10 trials / mice; Mann-Whitney test, two-tailed). **h**, Time percentage for pushing, voluntary retreating, resistance (when pushed by an opponent), and passive retreating (when pushed by an opponent) during tube tests in microbiota-depleted versus microbiota-intact mice before and after forced loss (Forced loss, $n = 13$ mice, 39 trials; Forced loss + ABX, $n = 7$ mice, 21 trials; Mann-Whitney test, two-tailed). Data are presented as mean \pm SEM. * $p < 0.05$, ** $p < 0.01$, *** $p < 0.001$, **** $p < 0.0001$; ns, not significant.

Comment 2.2: *Moreover, methodological details are occasionally unclear, such as how significant ASVs were identified. These aspects should be clarified in the text.*

Response 2.2: Following this reviewer's suggestion, we have revised the methods of our manuscript, especially the identification of ASVs in 16S rRNA gene sequence analysis (line 508-512; 517; 840-841). DNA sequences within microbiome are measured and analyzed to calculate their differences and similarities. Similar DNA sequences are clustered into the same amplicon sequence variants (ASVs), thereby obtaining an ASV number for each microorganism. Wilcoxon rank-sum test was used to compare the abundance of ASV between the two groups.

Additional Concerns and Suggestions

Comment 2.3: *Figure 1: The bar plot representing mean \pm SEM is not appropriate given that the authors used a non-parametric test due to the non-normal distribution of the data. I recommend replacing it with boxplots, which are better suited for such data.*

Response 2.3: We appreciate Reviewer #2's careful reading and suggestions. The bar plot in Fig. 1f is used to represent the time percentage of behaviors. During the competition of the tube test, dominant rank 1 mice mostly exhibited pushing and resisting behaviors when confronting subordinate mice, with little retreat behavior. Therefore, there were a

large number of zero-value data points in the retreat behavior. When we converted the time percentages of behaviors into box plots, the presence of many zero-value data points resulted in box plots without visible boxes, failing to effectively display the distribution of the data. After careful consideration, we have decided to use bar plots to present these data. Bar plots more clearly and intuitively display the data, especially in cases where there are many zero-value data points.

Response Figure 3

Fig. R3 Forced loss altered behaviors in mice. Time percentage of voluntary retreating and passive retreating (when pushed by an opponent) during tube test for mice before and after go-through-tube or forced loss process. The percentage of zero-value data in each group has been shown in the figure (Go-through-tube, $n = 7$; Forced loss, $n = 7$; 21 trials per group).

Comment 2.4: Line 122: The lack of differences in microbial community structure (Fig. S1) may partially be attributed to the coprophagic behavior of mice. This should be acknowledged here or discussed later in the manuscript.

Response 2.4: Thanks for your question. Coprophagy refers to the behavior of animals consuming their own and their cage mates' feces. Compared to non-coprophagic mice, coprophagic mice exhibited significant differences in their small intestinal microbiota. Moreover, the microbial community within the small intestine of non-coprophagic mice was more similar to that of healthy humans¹³. Microbiome assessments have shown that the gut microbial diversity was reduced in non-coprophagic mice, while coprophagy normalized the microbiota of co-housed mice^{10,11}. Therefore, coprophagic behavior can influence the gut microbiota structure of co-housed mice. In future studies, it will be necessary to control for this confounding factor in mouse model studies to address this issue. We have already elucidated the impact of coprophagic behavior on microbial structure in the discussion (line 396-399).

Comment 2.5: Figure 2C: The heatmap does not show the relative abundance of genera as stated in the figure legend but instead displays fold changes and correlations between genera and behaviors. Additionally, it is unclear whether the x-axis represents multiple ASVs per genus. If so, these should be reported together with their assigned taxa for clarity.

Response 2.5: Thank you for your questions and valuable comments. In the resubmitted

manuscript, we have added a legend to Figure 2c indicating the relative abundance of differential ASVs. Additionally, we have added the corresponding multiple ASVs for each genus in Figure 2c to ensure a clear and intuitive presentation of the figure.

Response Figure 4

Fig. R4 Abundance of differential ASVs and their correlation with behaviors in mice. Heatmap showed the relative abundance of differential ASVs across groups and their Spearman correlation with behaviors related to forced loss, with genus names of ASVs annotated (Wilcoxon rank-sum test, two-tailed).

Comment 2.6: *The method used to identify significantly different ASVs is unclear. Which test or tool was employed? This should be explicitly stated in the results, figure legends, and methods section.*

Response 2.6: Thank you for your careful reading. We used the Wilcoxon rank-sum test method to identify significantly different ASVs. We have added this part in the legend of Figure 2c and in the 16S rRNA gene sequence analysis methods of the manuscript (line 517; 840-841).

Comment 2.7: *Figure 2D: The correlations among ASVs in the two clusters could result from the compositional nature of microbiome data. This limitation should be acknowledged here and in the discussion. Any conclusions drawn from these*

correlations should ideally be tested experimentally.

Response 2.7: This reviewer poses an insightful point. We agree that the gut microbiota itself possesses a complex compositional nature. Numerous studies have revealed microbial potential interactions based on calculation of their composition ratio^{14,15}. In our study, we identified 16 differential ASVs associated with forced loss and created a network to display their correlations. However, this does not necessarily indicate that these correlations were caused by forced loss. Further studies, including in vitro co-culture experiments and functional metabolomics of microbial interactions, are required to elucidate the underlying interaction mechanisms. This limitation has been acknowledged and explained in the discussion (line 399-403).

Comment 2.8: Figure 2E: The LEfSe analysis appears forced to yield significant results. By default, LEfSe uses an LDA score of 2 as a threshold. Additionally, LEfSe applies a Wilcoxon test and reports nominal p-values without accounting for multiple comparisons. I suggest first testing for significant differences using a Wilcoxon rank-sum test corrected for multiple comparisons and then applying LEfSe on the significantly different pathways to identify the most biologically relevant ones, aligned with the hypothesis.

Response 2.8: We thank reviewer for these suggestions and revised our Figure 2E. We used the Wilcoxon rank-sum test with FDR correction for multiple comparisons to examine inter-group differences, which led to the identification of six differential functions presented in Figure 2e. LEfSe analysis illustrated the impact of differential functions contributing to inter-group differences, with LDA scores > 1.3 ¹⁶. The gray columns represented functions with LDA scores < 2 , and only butanoate metabolism with LDA score > 2 was enriched in the forced loss group (line 845-848).

Reference:

1. Wang, Z. *et al.* Gut microbiota regulate insomnia-like behaviors via gut-brain metabolic axis. *Mol Psychiatry* (2024) doi:10.1038/s41380-024-02867-0.
2. Li, Y. *et al.* Sodium butyrate alleviates lead-induced neuroinflammation and improves cognitive and memory impairment through the ACSS2/H3K9ac/BDNF pathway. *Environment International* **184**, 108479 (2024).
3. Zhou, X. *et al.* Gut microbiota dysbiosis in hyperuricaemia promotes renal injury through the activation of NLRP3 inflammasome. *Microbiome* **12**, 109 (2024).
4. Newhouse, P. & Albert, K. Estrogen, Stress, and Depression: A Neurocognitive Model. *JAMA Psychiatry* **72**, 727–729 (2015).
5. Albert, K. M. & Newhouse, P. A. Estrogen, Stress, and Depression: Cognitive and Biological Interactions. *Annu Rev Clin Psychol* **15**, 399–423 (2019).
6. Williamson, C. M. *et al.* Social hierarchy position in female mice is associated with plasma corticosterone levels and hypothalamic gene expression. *Sci Rep* **9**, 7324 (2019).
7. Fan, Z. *et al.* Neural mechanism underlying depressive-like state associated with social status loss. *Cell* **186**, 560-576.e17 (2023).
8. Choi, T.-Y. *et al.* Distinct prefrontal projection activity and transcriptional state

- conversely orchestrate social competition and hierarchy. *Neuron* **112**, 611-627.e8 (2024).
9. Noh, K. *et al.* Cortical astrocytes modulate dominance behavior in male mice by regulating synaptic excitatory and inhibitory balance. *Nat Neurosci* **26**, 1541–1554 (2023).
 10. Khuat, L. T. *et al.* Mechanisms by Which Obesity Promotes Acute Graft-Versus-Host Disease in Mice. *Front Immunol* **12**, 752484 (2021).
 11. Ericsson, A. C. & Franklin, C. L. Manipulating the Gut Microbiota: Methods and Challenges. *ILAR J* **56**, 205–217 (2015).
 12. Desjardins, C., Maruniak, J. A. & Bronson, F. H. Social Rank in House Mice: Differentiation Revealed by Ultraviolet Visualization of Urinary Marking Patterns. *Science* **182**, 939–941 (1973).
 13. Bogatyrev, S. R., Rolando, J. C. & Ismagilov, R. F. Self-reinoculation with fecal flora changes microbiota density and composition leading to an altered bile-acid profile in the mouse small intestine. *Microbiome* **8**, 19 (2020).
 14. Wang, S. *et al.* Microbial collaborations and conflicts: unraveling interactions in the gut ecosystem. *Gut Microbes* **16**, 2296603.
 15. Dai, Q. *et al.* Beyond bacteria: Reconstructing microorganism connections and deciphering the predicted mutualisms in mammalian gut metagenomes. *Ecol Evol* **13**, e9829 (2023).
 16. Hussan, H. *et al.* Distinctive patterns of sulfide- and butyrate-metabolizing bacteria after bariatric surgery: potential implications for colorectal cancer risk. *Gut Microbes* **15**, 2255345.